# Dynamics Modeling and Characterization of Sunk Screw Connection Structure in Small Rockets

**Xiaotian Zhang** [1,*] **, Ruiqing Wang** [1]**, Xiaogang Li** [2]**, Chengyang Lu** [1]**, Zhengkang Wang** [2] **and Wenlong Wang** [3]

1   School of Astronautics, Beihang University, Beijing 100191, China
2   Science and Technology on Space Physics Laboratory, Beijing 100076, China
3   Beijing Institute of Spacecraft System Engineering, Beijing 100094, China
*   Correspondence: zhangxiaotian@buaa.edu.cn

**Abstract:** Bolted flange joints are widely used in engineering structures. Sunk screw connection structures commonly used in small rockets and missiles exhibit significant nonlinear characteristics when subjected to forces. In this article, a study of the dynamic characteristics of sunk screw connection is conducted. A 3-dof trilinear dynamic model is proposed, based on the study of the stiffness characteristics of the connection structure and considering contact nonlinearities. The connection surface is simplified as two axial trilinear springs and a lateral linear spring. The motion of the system can be divided into nine regions by the turning point of the trilinear springs. So that the motion of the system in each region can be completely resolved, the dynamic characteristics of the 3-dof trilinear system under impulse load and simple harmonic load are studied by means of semi-numerical analytical method. It is found that the response frequency of the system remains unchanged under a small impulse load, and the response can be obtained by approximate analytical expressions. When the impulse load is large, the response frequency is fluctuant, which reflects the sensitivity of the nonlinear system to the magnitude of impulse load. Under the simple harmonic excitation of bending moment, the response frequency curve of the system presents good single peak characteristics when the excitation amplitude is small. When the amplitude is large, the peak frequency of the system shifts, and the phenomenon of multi-peak resonance is shown in a certain range.

**Keywords:** bolted joint; dynamic modeling; nonlinear system; dynamics characteristics

## 1. Introduction

The structure subsystem is an important part of the aircraft, which serves to connect the subsystems, protect the internal equipment, and withstand the static and dynamic loads [1] so that the rocket can maintain a good aerodynamic shape [2]. The increasing demand for small satellites has led to the continuous development of small rocket technology. The diameter of small rockets is small and the structure of the cabin is compact, so a sunk screw structure with high space utilization is often used to connect different cabins [3]. It is even the best choice for multi-segment connections where design space is limited [4].

Many scholars have studied the static properties of bolt joint structures [5,6]. For the analysis of dynamic properties, the simplified models are connected by linear constraint relations, such as multi-point coupling, rigid connection, or linear spring connection [7]. The values of stiffness parameters depend on experience or a large number of tests, which makes it difficult to obtain accurate dynamic parameters in the scheme design stage and increases the complexity of design iterations. The nonlinear characteristics of actual systems are difficult to obtain from linear simplification [8–10]. In addition to the complexity of the dynamics problem, the complex characteristics of the spacecraft flight environment, with multiple conditions of random loading, causes the bolted surfaces to have a significant impact on the dynamic characteristics of the spacecraft. Due to the large number of

bolted connections in rockets and missiles, and their diverse forms and relatively complex structures, there is no unified dynamics model for analysis. How to extract a simplified dynamics model to simulate the dynamics characteristics of bolted flange connection structures is an urgent problem for scholars to solve.

For nonlinear connection contact problems, scholars have proposed a series of constitutive models to describe their nonlinear mechanical properties [11,12]. These constitutive models can be used in kinetics simplification of complex systems, and the simplified models can reflect the system's stiffness properties and damping properties, to a certain extent. These constitutive models include the Bouc-Wen model [13], the Valanis model [14], the Jenkins element model [15], the Iwan model [16], the Dahl model [17], the LuGre model [18], the shear layer model [19,20], and the bilinear model [21].

In addition to the above models, some researchers have proposed a shear-layer model [22,23], which simplifies the joint surface into a thin layer that can withstand shear forces. Cigeroglu [24,25] proposed a one- and two-dimensional slip-movement friction model by replacing the rods in the shear-layer model with beams. Xiao [26] described the energy dissipation characteristics of a flat plate lap joint by considering the residual stiffness in the macroscopic slip phase based on the former two.

New simplified models of dynamics based on the force characteristics of the actual connection structure have been well applied in engineering. Nagata [27] performed a simplified modeling of the bolted flange connection structure by assuming the linearization of the gasket stress-strain characteristics. It was found that the bolt flange structure had different stiffness characteristics in tension and compression, and could be simplified as a bilinear spring. On this basis, Lu [28] discussed the dynamics and coupling characteristics of the system under different excitation. Kashani [29] proposed a bi-linear hysteretic model to study dry friction in bolted and riveted mechanical joints.

This paper takes the sunk screw joint structure as the study subject and investigates the dynamics of the joint structure through the finite element method. Based on the stiffness characteristics of the sunk screw joint structure, a simplified dynamics model of the structure is proposed, which is called a three-degree-of-freedom trilinear dynamic model (subsequently referred to as 3-dof trilinear dynamic model), and the response of the system is analytically calculated through the region division method to solve the equations of motion in each motion region of the system. Based on this model, the system response under the impact load in axial, lateral, and bending directions, as well as under the simple harmonic excitation, is studied by the semi-numerical analytical method. The nonlinear dynamic characteristics of the system are analyzed. This paper proposes an empirical formula for stiffness calculation with a certain range of applicability, which can be used to quickly calculate the joint surface stiffness at the early stage of rocket scheme design. Based on the stiffness study, a 3-dof trilinear dynamic modeling model is proposed to study the dynamics of the countersunk screw socket structure. Using the 3-dof trilinear dynamic modeling technique, the movement of the system in each area can be fully resolved and accurate joint structure dynamics can be obtained. Redundant or iterative designs due to lack of model accuracy in the initial design phase can be avoided. It provides a basis for the acquisition of dynamics modeling parameters and the design of the dynamic characteristics at the early stage of the rocket scheme design.

## 2. Simplified Dynamics Modeling of the Connection Structure

Extracting the mechanical characteristics of the sunk screw connection structure is beneficial for establishing a complete nonlinear rocket dynamics model.

### 2.1. Structural Stiffness of Sunk Screw Connection

A typical sunk screw connection structure is shown in Figure 1, including the outer flange, the inner flange, and the sunk screws uniformly distributed along the circumference. The dimensions of the connection structure in this section are shown in Table 1, defining the *X*-axis as axial and the *Y*-axis as lateral. To avoid excessive restraint, there is usually

an axial assembly gap Δ between the inner and outer flanges after assembly, and the sunk screw connection structure exhibits different pull-pressure characteristics when subjected to axial (*X*-direction) pressure or tension. In the axial direction of the rocket, the sunk screw connection structure can be simplified as a trilinear spring with different tensile and compressive properties. Its tensile and clearance compressive stiffnesses are mainly influenced by the screw stiffness, and gapless compressive stiffness is mainly influenced by the stiffness of the flange barrel section.

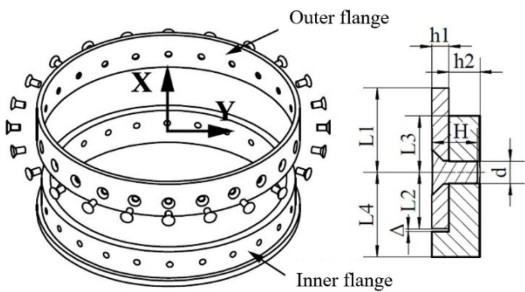

**Figure 1.** Cross section diagram of sunk screw connection structure and joint.

**Table 1.** Typical size of sunk screw connection structure.

| Parameter | *d* | *H* | *h1* | *h2* | *L1* | *L2* | *L3* | *L4* | Δ |
|---|---|---|---|---|---|---|---|---|---|
| size/mm | 8 | 16 | 6 | 11 | 30 | 20 | 20 | 30 | 0.2 |

The axial stiffness of the sunk screw connection structure can be divided into three stages:

$$k_s^* = \begin{cases} k_{s+} , \delta \geq 0 \\ k_{s0} , \Delta < \delta < 0 \\ k_{s-} , \delta \leq \Delta \end{cases} \tag{1}$$

The tensile stiffness is $k_{s+}$, the compression stiffness with clearance is $k_{s0}$, and the compression stiffness without clearance is $k_{s-}$. $\delta$ is the deformation, $\delta \geq 0$ means the connection structure is in tension, while $\Delta < \delta < 0$ means the connection structure is in compression and the displacement is less than the assembly gap. When the structure is in compression and the displacement is greater than the assembly gap, $\delta \leq \Delta$, here $\Delta$ is the negative number, which means opposite to the *X* direction, and its absolute value is the actual gap width.

The stiffness values of each linear segment can be obtained by FEA or experimental load-displacement curve fitting, but the stiffness values satisfying the accuracy requirements can also be obtained by simplified analytical algorithms at the early stage of the design of the rocket structure. For a single sunk screw joint, the main source of deformation in the tensile and gap-compression sections is the bending of the screw, and the main generator of the gap-less compression deformation is the compression of the cylindrical section of the inner and outer flange. The tensile section and the gapless compression section have different stiffness due to the different force states of the inner and outer flanges and the inconsistent thickness of the compressed end of the flange.

The sunk screw structure studied in this chapter consists of 24 countersunk screw joints (Figure 1), and a single countersunk screw joint is a substructure of the socket structure. The thickness of the body of cabin 1 is 2 mm, the thickness of the joint section is 6 mm, and the length is 304 mm. The thickness of cabin 2 is 11 mm, and the length is 100 mm. The diameter of the countersunk screw is 8 mm, and the length is 16 mm.

The 3D finite element modeling of the countersunk screw socket structure was performed by a commercial software, Hypermesh. Figure 2 shows the finite element model of the countersunk screw socket structure using eight-node hexahedron elements (HEX8). In order to reduce the computational scale, only half of the sunk screw structure was modeled

by imposing symmetry conditions, and only half of the corresponding loads were applied. The cantilever beam was modeled by a rigid element with one end connected to all nodes at the right end of cabin-2. Contact interface 1 was set between cabin-1 and sunk screw. Interface 2 was set between cabin-1 and cabin-2. Interface 3 was set between cabin-2 and sunk screw. Interface 1 and interface 2 were both in standard contact mode (contactable, separable), which set sliding friction coefficient as 0.2; interface 3 was bound contact. The material of each part was set to 30CrMnSi. More details and verification can be seen in the reference [30].

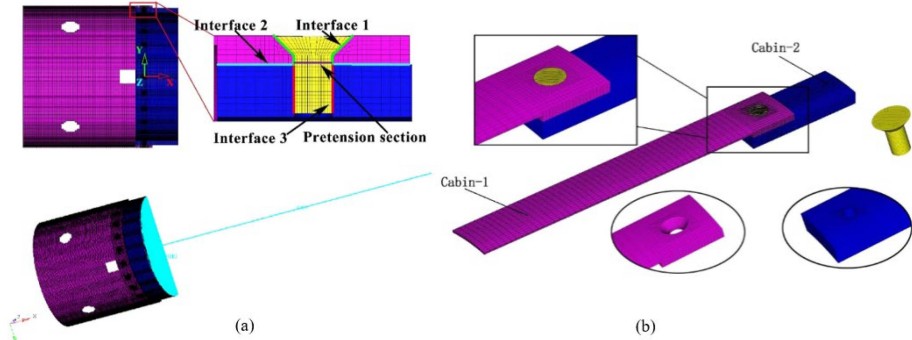

**Figure 2.** FE model of whole model (**a**) and substructure (**b**) of sunk screw connection.

### 2.1.1. Axial Tensile Stiffness

The static finite element analysis of the joint structure was carried out in ANSYS. The *X*-directional tension was loaded on cabin 1, and the three load conditions were 60 kN, 80 kN, and 120 kN. The deformation state of the joint section in the tensile state is shown in Figure 3. It can be seen that the inner and outer flanges basically generated displacement in the *X*-direction together, which means that the deformation of the flange itself was small.

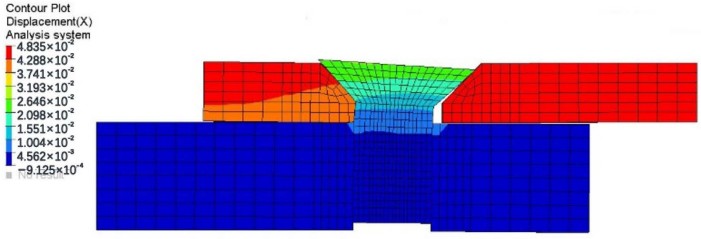

**Figure 3.** X-displacement of joint cross-section under tensile load (unit: mm).

From the displacement vector diagram of the sunk screw in the *X* direction in Figure 3, it can be seen that the sunk screw bends under the load, and the left part of the sunk head produces positive displacement in the *X* direction and the *Y* direction, and the displacement in the two directions is approximately equal in size in this example. The displacement in the *X* direction of the sunk part causes the outer flange moving in the *X* direction to follow the sunk screw; the displacement in the *Y* direction makes the left side of the sunk hole of the outer flange closer to the sunk screw, and also causes the displacement in the *X* direction. The combined effect of the two displacements is the deformation of the connection structure in the *X* direction.

As the sunk screw force and deformation state is similar to the cantilever beam, in the calculation of sunk screw deformation, we can refer to the cantilever beam deflection formula, with additional correction factors. The deformation of sunk screw in *X* direction is:

$$\delta_{sX} = a_{sX} \frac{FL^3}{3nEI} = a_{sX} \frac{64FL^3}{3\pi nEd^4} \tag{2}$$

The cantilever length $L$ is equal to the thickness of the outer flange, $E$ is taken as 200 GPa, $d$ is the screw diameter, the value of correction factor $a_{sX}$ is based on the finite element calculation results, $n$ is the number of sunk screws.

The deformation of sunk screws in the $Y$ direction is linearly related to the $X$ direction:

$$\delta_{sY} = a_{sY}\delta_{sX} \tag{3}$$

The deformations and correction factors of sunk screws under different tensile loads are shown in Table 2. The axial load in the table is the load on the connection structure, and the subscript $F$ of deformation indicates the finite element calculation result.

**Table 2.** Sunk screw displacement and correction factor (mm).

| Axial Load F | $\delta_{sX-F}$ | $\delta_{sY-F}$ | $a_{sX}$ | $\delta_{sX}$ | $a_{sY}$ | $\delta_{sY}$ |
|---|---|---|---|---|---|---|
| 60 kN | 0.01334 | 0.01323 | 3 | 0.01343 | 0.98 | 0.01316 |
| 120 kN | 0.02651 | 0.02591 | 3 | 0.02686 | 0.98 | 0.02632 |
| 180 kN | 0.03972 | 0.03870 | 3 | 0.04029 | 0.98 | 0.03948 |

Under the axial tensile load, the outer flange and the inner flange are locally compressed, and most of the area is in tension. In order to facilitate the analytical calculation, it can be assumed that the inner and outer flanges are uniformly stretched, and its actual deformation is obtained by multiplying the correction factor; therefore, the deformation of the inner and outer flanges in the $X$ direction is:

$$\delta_{fw} = a_{fw}\frac{4F(L_1 + L_2)}{\pi E_f(D_1^2 - D_2^2)} \tag{4}$$

$$\delta_{fn} = a_{fn}\frac{4F(L_3 + L_4)}{\pi E_f(D_2^2 - D_3^2)} \tag{5}$$

$a_{fw}$ and $a_{fn}$ are the deformation correction coefficients of the outer and inner flanges, respectively; elastic modulus of the flange material $E_f$ is taken as 200 GPa in this case; $D_1$, $D_2$ and $D_3$ are the outer flange outer diameter of 336 mm, inner diameter of 324 mm, and inner flange inner diameter of 302 mm, respectively; $L_1$, $L_2$, $L_3$ and $L_4$ are shown in Table 1.

The deformation and correction coefficients of the inner and outer flange in the $X$ direction under different tensile loads are shown in Table 3; the subscript $F$ of deformation indicates the finite element calculation result.

**Table 3.** $X$ direction deformation of internal and external flange and correction coefficient.

| Axial Load F | $\delta_{fw-F}$/mm | $\delta_{fn-F}$/mm | $a_{fw}$ | $\delta_{fw}$/mm | $a_{fn}$ | $\delta_{fn}$/mm |
|---|---|---|---|---|---|---|
| 60 kN | 0.00315 | 0.00202 | 1.3 | 0.00313 | 1.5 | 0.00208 |
| 120 kN | 0.00626 | 0.00410 | 1.3 | 0.00627 | 1.5 | 0.00416 |
| 180 kN | 0.00937 | 0.00618 | 1.3 | 0.00940 | 1.5 | 0.00624 |

Considering the deformation of sunk screws, external flange, and internal flange deformation, we can determine the deformation and tensile stiffness of sunk screw connection structure as:

$$\delta = a_s(\delta_{sX} + \delta_{sY}) + \delta_{fw} + \delta_{fn} \tag{6}$$

$$k_{s+} = \frac{F}{\delta} = \frac{1}{a_s a_{sX}\frac{64L^3}{3\pi nEd^4}(1 + a_{sY}) + a_{fw}\frac{4(L_1+L_2)}{\pi E_f(D_1^2 - D_2^2)} + a_{fn}\frac{4(L_3+L_4)}{\pi E_f(D_2^2 - D_3^2)}} \tag{7}$$

where the correction factor of sunk screw deformation is 0.715, $a_{sX} = 3$, $a_{sY} = 0.98$, $a_{fw} = 1.3$, $a_{fn} = 1.5$. According to Equation (7), the tensile stiffness of the sunk screw connection structure can be obtained.

Under different tensile loads, the deformation and tensile stiffness of the sunk screw connection structure are shown in Table 4. It can be seen that in the tensile state, the deformation of the sunk screw connection structure is linearly related to the axial load.

**Table 4.** Deformation and stiffness of sunk screw connection structure.

| Axial Load F | $\delta_F$/mm | $k_{s+F}$/N·m$^{-1}$ | $\delta$/mm | $k_{s+}$/N·m$^{-1}$ |
|---|---|---|---|---|
| 60 kN | 0.02435 | $2.464 \times 10^9$ | 0.02422 | $2.477 \times 10^9$ |
| 120 kN | 0.04835 | $2.482 \times 10^9$ | 0.04845 | $2.477 \times 10^9$ |
| 180 kN | 0.07238 | $2.487 \times 10^9$ | 0.07268 | $2.477 \times 10^9$ |

### 2.1.2. Gapped Compression Stiffness

In the gapped compression section, the stress state of the sunk screw is similar to the tensile section, though in the opposite direction. Compared with Figure 4, the displacement of the screw bottom in $Y$ direction (Figure 5) is basically zero, indicating that under compression load, the inner flange is a stronger constraint, and the force state of the sunk screw is similar to that of the cantilever beam. The internal and external flange force state under compression is different from that under tensile load; in the gapped compression section, the internal and external flanges are compressed on one side only, and the free end is almost unstressed, where deformation on the stiffness of the connection structure is very small and can be ignored.

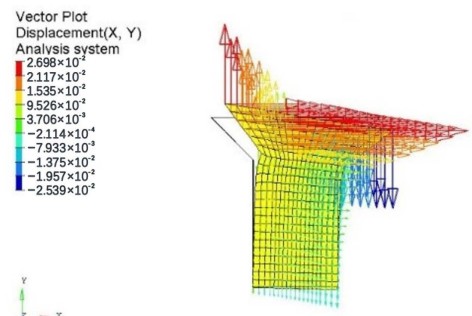

**Figure 4.** Displacement vector of screw cross-section in $X$ and $Y$ directions under tensile load (unit: mm).

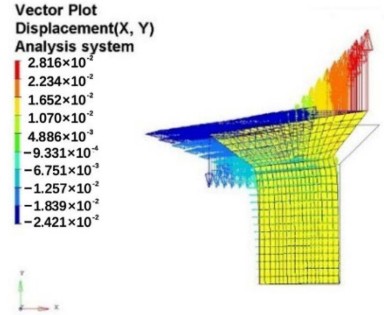

**Figure 5.** Sunk screw cross-section displacement vector diagram under compression load (unit: mm).

Under the compressive load, the $X$ and $Y$ direction deformation of sunk screws and the internal and external flange deformation are:

$$\delta_{sX} = a_{sX0}\frac{FL^3}{3nEI} = a_{sX0}\frac{64FL^3}{3\pi nEd^4} \tag{8}$$

$$\delta_{sY} = a_{sY0}\delta_{sX} \tag{9}$$

$$\delta_s = a_{s0}(\delta_{sX} + \delta_{sY}) \tag{10}$$

$$\delta_{fw} = a_{fw0} \frac{4FL_1}{\pi E_f (D_1^2 - D_2^2)} \tag{11}$$

$$\delta_{fn} = a_{fn0} \frac{4FL_4}{\pi E_f (D_2^2 - D_3^2)} \tag{12}$$

$$\delta = \delta_s + \delta_{fw} + \delta_{fn} \tag{13}$$

Then, the structural stiffness of the sunk screw connection in the gapped compression section can be expressed as:

$$k_{s0} = \frac{F}{\delta} = \frac{1}{a_{s0}a_{sX0}\frac{64FL^3}{3\pi nEd^4}(1+a_{sY0}) + a_{fw0}\frac{4FL_1}{\pi E_f (D_1^2 - D_2^2)} + a_{fn0}\frac{4FL_4}{\pi E_f (D_2^2 - D_3^2)}} \tag{14}$$

In the gapped compression section, the pressure load is 120 kN. The results of the finite element calculation of the deformation of each part, and the values of the correction factors, are shown in Table A1.

### 2.1.3. Gapless Compression Stiffness

When the axial pressure load causes the compression of the connection structure to be greater than the assembly gap, the two cabins are in contact with each other; then, the compression stiffness should be the gap-free compression stiffness plus the cylinder section stiffness.

$$k_{s-} = k_{s0} + \frac{E_f \pi (D_1^2 - D_2^2)}{4(L_1 + L_4)} \tag{15}$$

Substitute the parameters of the connection structure in it, we can get $k_{s-} = 2.357 \times 10^{10}$ N/m.

### 2.1.4. Lateral Stiffness

When the sunk screw connection structure is subjected to lateral (along the $Y$-axis) load, its displacement is shown in Figure 6, and the connection structure is tightly connected under the action of sunk screw. The force state is similar to a cantilever beam, with an approximate non-equal-thickness circular cross-section, as shown in the structure in the red wireframe in Figure 6. The circular thickness is, in sequence, the thickness of the inner flange $h_2$, the sum of the thickness of the inner and outer flange $h_1 + h_2$, and the thickness of the outer flange $h_1$.

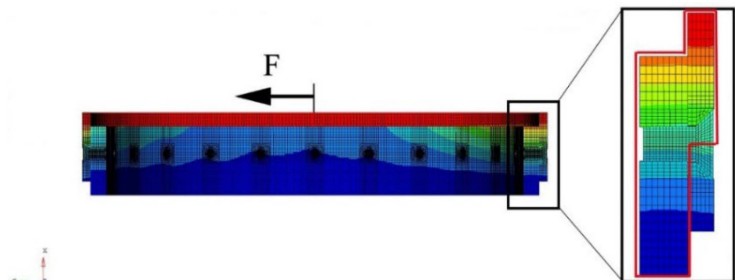

**Figure 6.** Displacement contour of sunk screw connection structure under lateral load.

According to the cantilever beam deflection formula, the lateral deformation of the connection structure is calculated as follows:

$$\delta_{r1} = \frac{F(L_1 - L_3)^3}{3E_f I_w} + \frac{F(L_1 - L_3)L_3^2}{2E_f I_{nw}} + \frac{FL_3^3}{3E_f I_{nw}} + \frac{FL_1 L_4^2}{2E_f I_n} + \frac{FL_4^3}{3E_f I_n} \tag{16}$$

The cross-sectional moment of inertia of the outer flange, the inner flange, and the combined inner and outer flange are in the following order:

$$I_w = \frac{\pi(D_1^4 - D_2^4)}{64} \tag{17}$$

$$I_n = \frac{\pi(D_2^4 - D_3^4)}{64} \tag{18}$$

$$I_{nw} = \frac{\pi(D_1^4 - D_3^4)}{64} \tag{19}$$

As the connection part is a short and thick-type structure, the shear effect needs to be considered, and the lateral deformation under shear force is:

$$\delta_{r2} = \frac{4FL_1}{\pi G_f(D_1^2 - D_2^2)} + \frac{4FL_4}{\pi G_f(D_2^2 - D_3^2)} \tag{20}$$

Total deformation is:

$$\delta_r = \delta_{r1} + \delta_{r2} \tag{21}$$

Adopting the correction factor, the lateral stiffness can be expressed as:

$$k_r = \frac{F}{a_r \delta_r} \tag{22}$$

The lateral stiffness can be obtained by finite element calculation $k_r = 1.95 \times 10^9 \text{N/m}$, the correction factor calculated from this result is $a_r = 4$.

### 2.2. Simplified Dynamic Model of Sunk Screw Connection Structure

By the study of the stiffness of the sunk screw connection structure, it is found that axial stiffness can be divided into three stages according to the installing gap between cabins, and the stiffness is linear in each region. Therefore, its axial stiffness can be equated to a trilinear spring (Equation (1)). Since the structure of the joint is tightly connected when laterally loaded under the action of sunk screws, its lateral stiffness can be equated to a linear spring. Based on this, the sunk screw connection structure can be simplified and a 3-dof trilinear dynamic model can be established to study the dynamics of the connection structure, and to lay the foundation for modeling the dynamics of the rocket.

### 2.2.1. 3-Dof Trilinear Dynamic Model

A simplified model of the dynamics of a typical sunk screw connection structure is shown in Figure 7, in which the axial ($X$-axis) spring is a trilinear spring with stiffness in the form shown in Equation (1), and the lateral ($Y$-axis) spring is a linear spring. The rectangular part represents the cabin structure, which has three directional degrees of freedom; the translational degrees of freedom in the $X$ and $Y$ directions $(u, v)$ and the rotational degrees of freedom $\theta$ around $O_c$. The mass and rotational inertia of the cabin structure are $m$ and $J$.

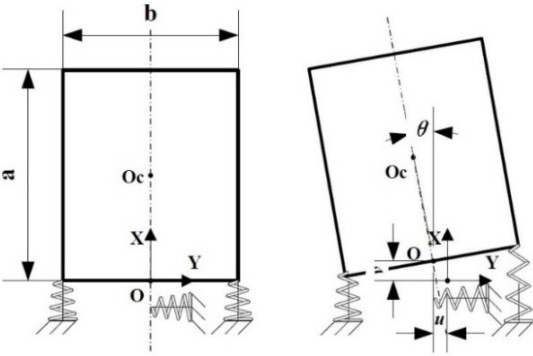

**Figure 7.** Initial and deformation state of 3-dof trilinear spring system.

Here, by decomposing the overall stiffness of the connection structure into two trilinear springs, the stiffness of each trilinear spring is half of the overall stiffness, and the dimension $b$ is not equal to the outside diameter of the pod, but $\sqrt{2}/2$ times that of the outside diameter. Assuming that the cabin segment of diameter $d$ is connected by $N$ uniformly distributed screws, and that the local stiffness of each screw joint is $k$, the overall stiffness of the connected structure is $Nk$. Simplifying this connected section to two springs of stiffness $k_0$, the springs are symmetrically distributed with a spacing $b$. To ensure that the equivalent model has the same effect as the original model under axial tension, it should be ensured that:

$$k_0 = Nk/2 \tag{23}$$

At the same time, it should be ensured that the effect of the moment load is the same. When the bending moment is $M$, the angle of rotation is $\theta$.

$$\begin{cases} M = 2k_0 \cdot \frac{b\theta}{2} \cdot \frac{b}{2} = \frac{k_0 \theta b^2}{2} \\ M = \sum\limits_{1}^{n} k \cdot \frac{\sin(2\frac{n}{N}\pi)d\theta}{2} \cdot \frac{\sin(2\frac{n}{N}\pi)d}{2} = \frac{Nk\theta d^2}{8} \end{cases} \tag{24}$$

Combining Equation (23) and Equation (24) yields

$$b = \frac{\sqrt{2}}{2}d \tag{25}$$

In order to solve the equations of motion of the module, let the displacement of the center of the bottom of the cabin be $x = (u, v, \theta)^T$, and the displacement of the midpoint of the bottom of the cabin and the center of mass of the cabin are as follows:

$$\begin{cases} u_{oc} = u - r\sin\theta \\ v_{oc} = v + r\cos\theta \\ \theta_{oc} = \theta \end{cases} \tag{26}$$

where $r$ is the height of the center of mass of the cabin, the velocity relation is obtained after differentiation, with the assumption of small deformation applied:

$$\begin{cases} \dot{u}_{oc} = \dot{u} - r\dot{\theta}\cos\theta = \dot{u} - r\dot{\theta} \\ \dot{v}_{oc} = \dot{v} + r\dot{\theta}\sin\theta = \dot{v} \\ \dot{\theta}_{oc} = \dot{\theta} \end{cases} \tag{27}$$

The system kinetic energy is:

$$T = \frac{1}{2}m\dot{u}_{oc}^2 + \frac{1}{2}m\dot{v}_{oc}^2 + \frac{1}{2}J\dot{\theta}_{oc}^2 \tag{28}$$

Substituting into the velocity relationship, Equation (27) yields:

$$T = \frac{1}{2}m\left(\dot{u} - r\dot{\theta}\right)^2 + \frac{1}{2}m\dot{v}^2 + \frac{1}{2}J\dot{\theta}^2 \tag{29}$$

The expression for the deformation of the spring is:

$$\begin{cases} \delta_1 = v - \frac{b}{2}\sin\theta \\ \delta_2 = v + \frac{b}{2}\sin\theta \\ \delta_3 = u \end{cases} \tag{30}$$

$\delta_1$ is the deformation of left axial spring, $\delta_2$ and $\delta_3$ are the deformation of right axial and radial spring. Since the axial spring is a trilinear spring, its elastic potential energy should be expressed in the following terms:

$$U_s^*(\delta) = \begin{cases} \frac{1}{2}k_{s+}\delta^2, \delta > 0 \\ \frac{1}{2}k_{s0}\delta^2, \Delta < \delta \le 0 \\ \frac{1}{2}k_{s0}\Delta^2 + \frac{1}{2}k_{s-}(\delta - \Delta)^2, \delta \le \Delta \end{cases} \tag{31}$$

where $\delta$ is the deformation of the trilinear spring, which can be $\delta_1$ or $\delta_2$, and $\Delta$ is the assembly gap of the connection structure. The elastic potential energy of the system is:

$$U = U_s^*(\delta_1) + U_s^*(\delta_2) + \frac{1}{2}k_r u^2 \tag{32}$$

The expression for the elastic potential energy of the system should be expressed in segments with 0 and $\Delta$ as the dividing point, in order to build a specific expression for the equation of motion of the system based on the Lagrange Equation.

2.2.2. Equations of Motion of the System in Each Region

In different regions, the springs are in different states of tension and compression, resulting in different stiffness matrices of the system. The system is divided into nine regions according to the tension and compression states of the springs on the left and right sides, as shown in Figure 8, and the motion of the system in each region can be determined.

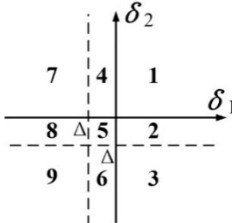

**Figure 8.** Partition diagram of 3-dof trilinear system.

(1) Region 1

When the system is moving in region 1, $\delta_1 > 0$ and $\delta_2 > 0$, both left and right springs, are stretched and the stiffness value is $k_{s+}$, then the elastic potential energy of the system can be expressed as:

$$U = U_s^*(\delta_1) + U_s^*(\delta_2) + \frac{1}{2}k_r u^2 = \frac{1}{2}k_{s+}\delta_1^2 + \frac{1}{2}k_{s+}\delta_2^2 + \frac{1}{2}k_r u^2 \tag{33}$$

The Lagrangian Equation of the system is:

$$\frac{d}{dt}\left(\frac{\partial T}{\partial \dot{q}_i}\right) - \frac{\partial T}{\partial q_i} + \frac{\partial U}{\partial q_i} = 0 \tag{34}$$

Substituting Equations (29) and (33) into Equation (34), and taking the generalized coordinates as $(u, v, \theta)$, the undamped free vibration equation of the system in region 1 is:

$$\begin{bmatrix} m & 0 & -mr \\ 0 & m & 0 \\ -mr & 0 & J+mr^2 \end{bmatrix} \begin{bmatrix} \ddot{u} \\ \ddot{v} \\ \ddot{\theta} \end{bmatrix} + \begin{bmatrix} k_r & 0 & 0 \\ 0 & 2k_{s+} & 0 \\ 0 & 0 & \frac{b^2}{2}k_{s+} \end{bmatrix} \begin{bmatrix} u \\ v \\ \theta \end{bmatrix} = 0 \tag{35}$$

The characteristic equation of the system in region 1 is:

$$\left| \begin{bmatrix} k_r & 0 & 0 \\ 0 & 2k_{s+} & 0 \\ 0 & 0 & \frac{b^2}{2}k_{s+} \end{bmatrix} - \omega^2 \begin{bmatrix} m & 0 & -mr \\ 0 & m & 0 \\ -mr & 0 & J+mr^2 \end{bmatrix} \right| = 0 \tag{36}$$

(2) Region 2

When moving in region 2, the equation of motion of the system is:

$$\begin{bmatrix} m & 0 & -mr \\ 0 & m & 0 \\ -mr & 0 & J+mr^2 \end{bmatrix}\begin{bmatrix} \ddot{u} \\ \ddot{v} \\ \ddot{\theta} \end{bmatrix} + \begin{bmatrix} k_r & 0 & 0 \\ 0 & k_{s+}+k_{s0} & \frac{b}{2}(k_{s0}-k_{s+}) \\ 0 & \frac{b}{2}(k_{s0}-k_{s+}) & \frac{b^2}{4}(k_{s+}+k_{s0}) \end{bmatrix}\begin{bmatrix} u \\ v \\ \theta \end{bmatrix} = 0 \qquad (37)$$

The characteristic equation of the system in region 2 is:

$$\left| \begin{bmatrix} k_r & 0 & 0 \\ 0 & k_{s+}+k_{s0} & \frac{b}{2}(k_{s0}-k_{s+}) \\ 0 & \frac{b}{2}(k_{s0}-k_{s+}) & \frac{b^2}{4}(k_{s+}+k_{s0}) \end{bmatrix} - \omega^2 \begin{bmatrix} m & 0 & -mr \\ 0 & m & 0 \\ -mr & 0 & J+mr^2 \end{bmatrix} \right| = 0 \qquad (38)$$

(3) Region 3

When moving in region 3, $\delta_1 > 0$ and $\delta_2 < \Delta$, the left spring is in tension with a stiffness value of $k_{s+}$, and the right spring is in gapless compression with a stiffness value of $k_{s-}$, then the elastic potential energy of the system can be expressed as:

$$U = U_s^*(\delta_1) + U_s^*(\delta_2) + \frac{1}{2}k_r u^2 = \frac{1}{2}k_{s+}\delta_1^2 + \frac{1}{2}k_{s0}\Delta^2 + \frac{1}{2}k_{s-}(\delta_2-\Delta)^2 + \frac{1}{2}k_r u^2 \qquad (39)$$

Substituting the kinetic and potential energies of the system, Equations (29) and (39), into Equation (34) and taking the generalized coordinates as $(u, v, \theta)$, the equation of motion of the system in region 3 is:

$$\begin{bmatrix} m & 0 & -mr \\ 0 & m & 0 \\ -mr & 0 & J+mr^2 \end{bmatrix}\begin{bmatrix} \ddot{u} \\ \ddot{v} \\ \ddot{\theta} \end{bmatrix} + \begin{bmatrix} k_r & 0 & 0 \\ 0 & k_{s+}+k_{s-} & \frac{b}{2}(k_{s-}-k_{s+}) \\ 0 & \frac{b}{2}(k_{s-}-k_{s+}) & \frac{b^2}{4}(k_{s+}+k_{s-}) \end{bmatrix}\begin{bmatrix} u \\ v \\ \theta \end{bmatrix} = \begin{bmatrix} 0 \\ \Delta(k_{s-}-k_{s0}) \\ \frac{b}{2}\Delta(k_{s-}-k_{s0}) \end{bmatrix} \qquad (40)$$

It can be seen that, although the system is not subject to external forces, the equation of motion is in the form of forced vibration due to the segmented linearity of the spring, which causes the system to enter the gapless compression state motion with additional external force effects. The equation of motion is in the form of forced vibration.

The characteristic equation of motion of the system in region 3 is:

$$\left| \begin{bmatrix} k_r & 0 & 0 \\ 0 & k_{s+}+k_{s-} & \frac{b}{2}(k_{s-}-k_{s+}) \\ 0 & \frac{b}{2}(k_{s-}-k_{s+}) & \frac{b^2}{4}(k_{s+}+k_{s-}) \end{bmatrix} - \omega^2 \begin{bmatrix} m & 0 & -mr \\ 0 & m & 0 \\ -mr & 0 & J+mr^2 \end{bmatrix} \right| = 0 \qquad (41)$$

(4) Region 4–9

Similarly, the characteristic equations of motion of the system in other regions can be obtained.

At this point, the motion of the system in each region is completely determined, and the equations of motion in each region can be unified and expressed as:

$$\begin{bmatrix} m & 0 & -mr \\ 0 & m & 0 \\ -mr & 0 & J+mr^2 \end{bmatrix}\begin{bmatrix} \ddot{u} \\ \ddot{v} \\ \ddot{\theta} \end{bmatrix} + \begin{bmatrix} k_{11} & 0 & 0 \\ 0 & k_{22} & k_{23} \\ 0 & k_{32} & k_{33} \end{bmatrix}\begin{bmatrix} u \\ v \\ \theta \end{bmatrix} = Q^* \qquad (42)$$

$$\begin{cases} k_{11} = k_r \\ k_{22} = k_{s1}^* + k_{s2}^* \\ k_{33} = \frac{b^2}{4}\left(k_{s1}^* + k_{s2}^*\right) \\ k_{23} = k_{32} = \frac{b}{2}\left(k_{s2}^* - k_{s1}^*\right) \end{cases} \qquad (43)$$

$k_{s1}^*$ and $k_{s2}^*$ are the stiffnesses of the left and right springs, whose values are determined by the partition in which the system is located; $Q^*$ is the additional generalized external forces on the system when the left or right spring is in the gapless compression section, the value of which is also determined by the partition in which the system is located. Let

$$K_{23} = \begin{bmatrix} k_{22} & k_{23} \\ k_{32} & k_{33} \end{bmatrix} \qquad (44)$$

The spring states, the stiffness matrix, and the additional generalized external forces in each region are shown in Table A2.

2.2.3. Response Analysis of the System within Each Region

(1) Free vibration region

When the motion of the 3-dof trilinear system is located in regions 1, 2, 4, and 5, the system moves as an undamped free vibration, shown below:

$$M\ddot{x} + Kx = 0 \tag{45}$$

The solution of the equation of motion within the region is (the derivation is shown in the Appendix A):

$$\mathbf{x}(t) = \sum_{i=1}^{n} \mathbf{A}^{(i)} \left( \frac{\mathbf{A}^{(i)T}\mathbf{M}\mathbf{x}(0)}{M_i} \cos \omega_i t + \frac{\mathbf{A}^{(i)T}\mathbf{M}\dot{\mathbf{x}}(0)}{\omega_i M_i} \sin \omega_i t \right) \tag{46}$$

According to the equations of motion and characteristic equation of each zone, the response of the system can be obtained by combining Equations (45) and (46) after substituting the initial conditions of the system.

(2) Forced vibration region

When the 3-dof trilinear system is located in regions 3, 6, 7, 8, and 9, the system is not subject to external forces, but, due to the nonlinearity of the system, the effect of additional external forces is produced; considering the damping of the classical system, the equation of motion is:

$$\mathbf{M\ddot{x} + C\dot{x} + Kx = Q} \tag{47}$$

After solving the matrix of the vibration modes of the undamped free vibration system, the displacement is expressed in the form of $n$ orthogonal principal vibration modes, according to the expansion theorem:

$$\mathbf{x}(t) = \sum_{i=1}^{n} \mathbf{A}^{(i)} y_i(t) = \mathbf{\Phi y}(t) \tag{48}$$

Substituting Equation (47) into Equation (46) and multiplying the left side of $\mathbf{\Phi}^T$ on both sides of the equation:

$$\mathbf{\Phi}^T \mathbf{M\Phi\ddot{y}}(t) + \mathbf{\Phi}^T \mathbf{C\Phi\dot{y}}(t) + \mathbf{\Phi}^T \mathbf{K\Phi y}(t) = \mathbf{\Phi}^T \mathbf{Q} \tag{49}$$

$$\begin{bmatrix} M_1 & 0 & \cdots & 0 \\ 0 & M_2 & \cdots & 0 \\ \vdots & \vdots & \ddots & \vdots \\ 0 & 0 & \cdots & M_n \end{bmatrix} \ddot{\mathbf{y}}(t) + \begin{bmatrix} C_1 & 0 & \cdots & 0 \\ 0 & C_2 & \cdots & 0 \\ \vdots & \vdots & \ddots & \vdots \\ 0 & 0 & \cdots & C_n \end{bmatrix} \dot{\mathbf{y}}(t) + \begin{bmatrix} K_1 & 0 & \cdots & 0 \\ 0 & K_2 & \cdots & 0 \\ \vdots & \vdots & \ddots & \vdots \\ 0 & 0 & \cdots & K_n \end{bmatrix} \mathbf{y}(t) = \begin{bmatrix} Q_1^* \\ Q_2^* \\ \vdots \\ Q_n^* \end{bmatrix} \tag{50}$$

It can be seen that the response solution problem for a multi-degree-of-freedom system has been converted into a response solution problem for a system with $n$ single degrees of freedom:

$$\ddot{y}_i(t) + 2\xi_i \omega_i \dot{y}_i(t) + \omega_i^2 y_i(t) = \frac{Q_i^*}{M_i} \quad (i = 1, 2, \cdots, n) \tag{51}$$

$$\xi_i = \frac{C_i}{2\omega_i M_i} \tag{52}$$

$$\omega_i = \sqrt{\frac{K_i}{M_i}} \tag{53}$$

Based on the Duhamel integral, solving for the viscous damped-forced vibration of a 1-dof system under arbitrary excitation yields:

$$y_i(t) = e^{-\xi_i\omega_i t}\left[y_i(0)\cos\omega_{Di}t + \frac{\dot{y}_i(0) + \xi_i\omega_i y_i(0)}{\omega_{Di}}\sin\omega_{Di}t\right] + \frac{1}{M_i\omega_{Di}}\int_0^t Q_i^* e^{-\xi_i\omega_i(t-\tau)}\sin\omega_{Di}(t-\tau)\mathrm{d}\tau \quad (54)$$

$$\omega_{Di} = \omega_i\sqrt{1 - \xi_i^2} \quad (55)$$

Let $t = 0$ in Equation (48) and left multiply on both sides of the Equation by $\mathbf{A}^{(i)T}\mathbf{M}$ to obtain

$$\mathbf{A}^{(i)T}\mathbf{M}\mathbf{x}(0) = \mathbf{A}^{(i)T}\mathbf{M}\sum_{j=1}^n \mathbf{A}^{(j)}y_j(0) \quad (56)$$

According to the orthogonality of the vibration pattern, it yields:

$$y_i(0) = \frac{\mathbf{A}^{(i)T}\mathbf{M}\mathbf{x}(0)}{M_i} \quad (57)$$

Similarly, taking the derivative of both sides of Equation (48) and left multiplying $\mathbf{A}^{(i)T}\mathbf{M}$ gives:

$$\dot{y}_i(0) = \frac{\mathbf{A}^{(i)T}\mathbf{M}\dot{\mathbf{x}}(0)}{M_i} \quad (58)$$

The response of a multi-degree-of-freedom system to forced vibration under classical damping conditions can be obtained by substituting Equation (54) into Equation (48).

## 3. Dynamic Characteristics of the Cabin Connection Structure

Based on the 3-dof trilinear dynamic equivalent model, the dynamic response and coupling characteristics of the sunk screw connection structure under different excitations are studied.

### 3.1. Half-Numerical Analysis Method

The 3-dof trilinear system is a nonlinear system, and its response is difficult to completely resolve. However, according to the tensile and compressive characteristics of the trilinear spring, the system can be divided into nine motion regions. In each region, the system response can be completely resolved, and the analysis results of the system response in each region have been given in the previous section. Therefore, during the motion of the system, it is only necessary to judge the motion partition in which the system is located, and calculate the moments of entering and leaving the partition to obtain the complete response process of the system. In this calculation process, the motion in the partition is obtained by analytic calculation, and the moments at the boundary of the partition are obtained by numerical iterative calculation; therefore, it is called the half-numerical analysis method.

The main calculation process of the half-numerical analysis method can be briefly described as follows.

1. Obtaining the analytical solution of the system response in each partition according to the system parameters, and inputting the external excitation and initial conditions of the system.
2. Determining the subzone in which the system is located and applying the analytical formula for that subzone to calculate the system state at the next time step and recording it.
3. After completing the calculation of each time step, determine whether the system crosses the region, and repeat the calculation in step 2 if it does not, or return to the previous time step if it does.
4. Calculate the cross-zone moment using the dichotomous method, and use the analytical formula to calculate the system state at that moment.
5. Repeat steps 2–4 until the solution time is reached.

### 3.2. Verification of Half-Numerical Analysis Method

In order to verify the accuracy of the semi-numerical analysis method, a corresponding finite element model was built in ANSYS-APDL to compare the system response obtained from both. The parameters of the validation model are shown in Table 5.

**Table 5.** System parameters of the verification model.

| Parameter | $M$/kg | $J$/kg·m$^2$ | $b$/m | $r$/m | $\Delta$/m |
|---|---|---|---|---|---|
| Value | 1000 | 338.4 | 2 | 1 | 0.0002 |
| Parameter | $k_{s+}$/N·m$^{-1}$ | $k_{s0}$/N·m$^{-1}$ | $k_{s-}$/N·m$^{-1}$ | $k_r$/N·m$^{-1}$ | |
| Value | $10^6$ | $1.2 \times 10^6$ | $2 \times 10^7$ | $10^7$ | |

In the half-numerical analysis method, the calculation time is chosen to be 0.5 s, the fixed time step is $10^{-3}$ s, and the calculation accuracy is $10^{-15}$ s. Since the cross-sectional area of the cabin segment is much larger than the screw, the stress and strain of the cabin segment can be negligible compared with the screw under the action of external forces, so the cabin is considered as a rigid body in the semi-numerical model to improve the calculation efficiency.

In the finite element model, the fixed time step is $10^{-4}$ s, and the numerical calculation attenuation factor is taken to be 0, which means no attenuation. In the Matlab calculation, the cabin section is assumed to be a rigid body, and in order to reduce the model error of the two calculation methods, the elastic modulus of the cabin material in the FE model is enlarged to one thousand times the normal value. Then, the system response obtained by the two methods is shown in Figure 9.

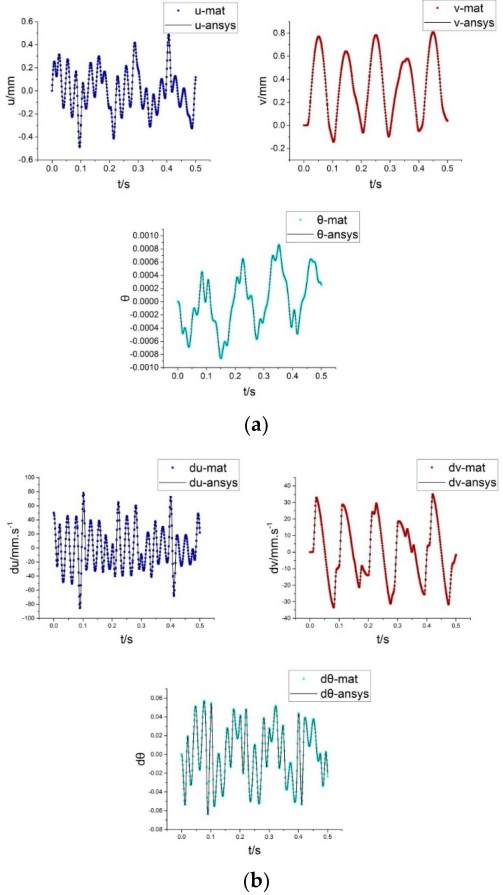

**Figure 9.** Comparison chart of system displacement (**a**) and velocity (**b**) response.

In Figure 9, the symbol "mat" represents the response obtained by the half-numerical analysis method, and the symbol "ansys" represents the finite element result. It can be seen that the displacement and velocity responses of the system obtained by the two methods are almost identical, which proves that the half-numerical analysis method can calculate the three-degree-of-freedom trilinear dynamical system with accurate results. Since the semi-numerical analytical calculation method is fully resolved in the partitioned motion response, its calculation is fast and accurate, and there is no convergence problem. Compared with the finite element method, the system response can be solved faster and more accurately when the equations of motion have been established, so it is suitable for studying the dynamics of the countersunk screw socket structure.

### 3.3. Response under Impact Loading of Sunk Screw Connection Structure

A typical sunk screw connection system is simplified to a 3-dof trilinear system, as shown in Table A3, and its dynamic characteristics are studied under different excitation forms and intensities. The response frequencies and vibration patterns in each division of the system are shown in Table A4.

#### 3.3.1. Response under Axial Impact Loading

For a 3-dof trilinear system, the system generates an initial velocity along the axial direction under an axial shock, ignoring the time effect of the shock load, and directly assigning the system initial conditions $x(0) = (0,0,0)$, $\dot{x}(0) = (0,\dot{v},0)$, where $\dot{v} \neq 0$, and the system response is calculated by the half-numerical analysis method.

Take $\dot{v} = 0.2 \sim 1$ m/s, the system only vibrates along the axial direction. Its axial displacement curve is shown in Figure 10, and dv is the axial initial velocity in the legend. The 0.1 s response time of the system is intercepted, and the motion time of each region of the system is counted, as shown in Table 6. Under the condition that the initial displacement of the system is zero, when the system is subjected to a small axial shock, the system only moves in region 1 and region 5, and the axial response frequency of the system is fixed. When the system is subjected to a large axial shock, the system moves in regions 1, 5, and 9, and the axial response frequency of the system increases with the increase in the shock load.

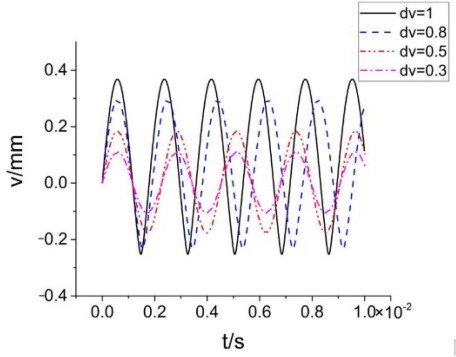

**Figure 10.** System displacement response under axial impact.

**Table 6.** Movement time of the system in different partitions under axial impact.

| $\dot{v}$/m·s$^{-1}$ | $t_1$/s | $t_5$/s | $t_9$/s |
|---|---|---|---|
| 0.2 | 0.0511 | 0.0489 | 0 |
| 0.5 | 0.0511 | 0.0489 | 0 |
| 0.8 | 0.0607 | 0.0289 | 0.0104 |
| 1.0 | 0.0648 | 0.0236 | 0.0116 |

Notes: The subscript of $t$ is the region number.

The first-order response frequency of the system is shown in Figure 11, conducting the fast Fourier transform. Under the condition that the initial displacement is zero, the axial response frequency of the system is 441 Hz when the initial axial velocity satisfies

$\dot{v} \leq 0.7$ m/s. Combined with Figure 10 and Table 6, it can be seen that the system only moves in regions 1 and 5 at this time, and its axial response frequency is independent of the magnitude of the impact load. With initial axial velocity $\dot{v} \geq 0.8$ m/s, the axial response frequency of the system increases with the increase of the impact load magnitude, which is due to the fact that the system starts to move in region 9.

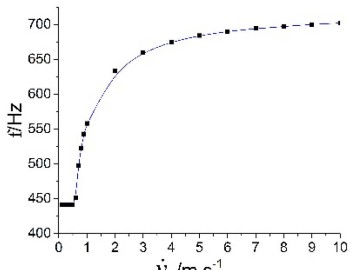

**Figure 11.** First-order response frequency of the system under axial shock.

In fact, by substituting the frequencies and vibration shapes of the system in Table A4 into Equations (46) and (48), we can learn that the system has only axial displacement response in region 1, region 5, and region 9, and its response frequencies are $f_{(1)} = 432.2$ Hz, $f_{(5)} = 450.2$ Hz and $f_{(9)} = 2179.3$ Hz. When the axial impact load is small, the system moves only in regions 1 and 5, and within one vibration cycle, the system moves in region 1 with time of $1/(2f_{(1)})$ and in region 5 with time of $1/(2f_{(5)})$. Let the system's motion frequency be $f$ and period $1/f$, and then, the theoretical response frequency of the system satisfies:

$$\frac{1}{f} = \frac{1}{2}\left(\frac{1}{f_{(1)}} + \frac{1}{f_{(5)}}\right) \tag{59}$$

Calculation by substitution gives $f = 441$ Hz, which is consistent with the fast Fourier transform results.

When the axial shock load is large, the system moves in regions 1, 5, and 9. Region 5 is the region formed by the assembly gap; when the axial shock is large enough, the system moves in region 5 for a short time, and this part of time can be ignored in the limit state, when the theoretical response frequency of the system satisfies:

$$\frac{1}{f} = \frac{1}{2}\left(\frac{1}{f_{(1)}} + \frac{1}{f_{(9)}}\right) \tag{60}$$

Then, we can get $f = 722.3$ Hz.

Since the actual shock load cannot be very large, region 5 still affects the response frequency of the system, the magnitude of which depends on the magnitude of the shock load, and the response frequency should be between 441 and 722.3 Hz. The system response is highly sensitive to the magnitude of the load, which is one of the typical characteristics of nonlinear systems.

### 3.3.2. Response under Lateral Impact Loading

Similar to the axial shock, for a 3-dof trilinear system, the system generates an initial velocity along the lateral direction under the lateral shock, neglecting the time effect of the shock load, and directly assigning the system initial conditions $\mathbf{x}(0) = (0, 0, 0), \dot{\mathbf{x}}(0) = (\dot{u}, 0, 0)$, where $\dot{u} \neq 0$; the system response is calculated by the half-numerical analysis method.

The displacement response of the system is shown in Figure 12, by taking $\dot{u} = 0.5$ m/s and $\dot{u} = 2$ m/s, respectively. When the initial displacement of the system is zero, the system will produce axial displacement and turning angle in bending direction under the lateral impact load, and the three directions of system motion are completely coupled. When the lateral impact load is small (e.g., $\dot{u} = 0.5$ m/s), the axial displacement of the system is not significant, and the main motion is rotation and translation along the lateral

direction; when the lateral impact load is large (e.g., $\dot{u} = 2$ m/s), it will produce a large axial displacement response, of which the magnitude is close to the lateral displacement.

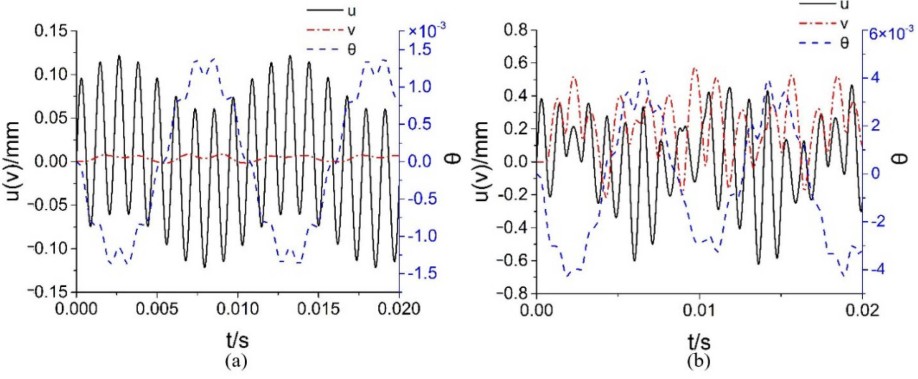

**Figure 12.** Displacement response with $\dot{u} = 0.5$ m/s (**a**) and $\dot{u} = 2$ m/s (**b**).

The displacement response of the system was intercepted within 1 s, and the motion time of the system in each division was counted, as shown in Table 7. The response is very sensitive to the magnitude of the impact load, and the first-order response frequencies of the system in each degree of freedom under different lateral impact conditions are shown in Figure 13, by performing the fast Fourier transform. It can be seen that the first-order response frequencies in the lateral and bending directions under the impact conditions are identical, while the first-order response frequency in the axial direction is twice as high as that in the lateral and bending directions in most cases. Due to the nonlinear characteristics of the system, the axial direction produces a higher first-order response frequency under certain impact conditions.

**Table 7.** Movement time of the system in each partition under lateral impact load.

| $\dot{u}$/m·s$^{-1}$ | $t_1$/s | $t_2$/s | $t_3$/s | $t_4$/s | $t_5$/s | $t_6$/s | $t_7$/s | $t_8$/s | $t_9$/s |
|---|---|---|---|---|---|---|---|---|---|
| 0.3 | 0.0002 | 0.4987 | 0 | 0.4976 | 0.0035 | 0 | 0 | 0 | 0 |
| 0.5 | 0.0002 | 0.4987 | 0 | 0.4976 | 0.0035 | 0 | 0 | 0 | 0 |
| 0.6 | 0.0014 | 0.4969 | 0.0007 | 0.4963 | 0.0041 | 0 | 0.0006 | 0 | 0 |
| 0.8 | 0.0631 | 0.3637 | 0.0611 | 0.3659 | 0.0840 | 0.0007 | 0.0607 | 0.0007 | 0 |
| 1.0 | 0.0589 | 0.3431 | 0.0874 | 0.3359 | 0.0782 | 0.0034 | 0.0902 | 0.0028 | 0 |
| 1.5 | 0.2170 | 0.2534 | 0.1071 | 0.2449 | 0.0460 | 0.0123 | 0.1035 | 0.0144 | 0.0015 |
| 2.0 | 0.5599 | 0.1093 | 0.0460 | 0.0897 | 0.0535 | 0.0410 | 0.0448 | 0.0353 | 0.0206 |

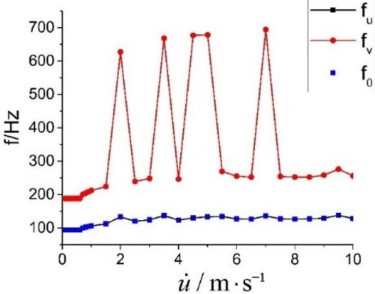

**Figure 13.** First-order response frequency of the system under lateral impact loading.

(1) System response under small lateral impact loads.

When the lateral impact load is small, the first-order response frequencies of the system in the lateral, axial, and corner directions are approximately 94 Hz, 188 Hz, and 94 Hz, respectively, and remain constant. Table 7 shows that the main motion regions of the system at this time are regions 2 and 4, and the spring states on the left and right sides are tensile on

one side and compressive on the other side, and the compression is less than the assembly gap $\Delta$. The tensile stiffness of the tri-linear spring is $k_{s+} = 2.95 \times 10^8$ N/m, and the compression stiffness with clearance is $k_{s0} = 3.2 \times 10^8$ N/m; the stiffness values are close to each other, so the symmetry of the system motion is good. Considering only the motion of the system in regions 2 and 4, and substituting the frequencies and vibration patterns of the two regions and the initial conditions of the system into Equation (46), the system response can be approximately resolved. With initial conditions of $\mathbf{x}(0) = (0,0,0)$, $\dot{\mathbf{x}}(0) = (\dot{u},0,0)$, the system response in region 2 is:

$$
\mathbf{x}_2(t) = \dot{u} \begin{pmatrix} [0.6336\sin(2\pi \times 94.1t) + 0.00044\sin(2\pi \times 441.3t) + 1.7983\sin(2\pi \times 849.4t)] \times 10^{-4} \\ [1.32\sin(2\pi \times 94.1t) - 0.3801\sin(2\pi \times 441.3t) + 0.0492\sin(2\pi \times 849.4t)] \times 10^{-5} \\ [-2.6\sin(2\pi \times 94.1t) + 0.0000068\sin(2\pi \times 441.3t) + 0.28692\sin(2\pi \times 849.4t)] \times 10^{-3} \end{pmatrix} \tag{61}
$$

When the system enters region 4 from region 2, the initial condition is approximated as $\mathbf{x}(0) = (0,0,0), \dot{\mathbf{x}}(0) = (-\dot{u},0,0)$; then, the response in region 4 is:

$$
\mathbf{x}_4(t) = -\dot{u} \begin{pmatrix} [0.6336\sin(2\pi \times 94.1t) + 0.00044\sin(2\pi \times 441.3t) + 1.7983\sin(2\pi \times 849.4t)] \times 10^{-4} \\ [-1.32\sin(2\pi \times 94.1t) + 0.3801\sin(2\pi \times 441.3t) - 0.0492\sin(2\pi \times 849.4t)] \times 10^{-5} \\ [-2.6\sin(2\pi \times 94.1t) + 0.0000068\sin(2\pi \times 441.3t) + 0.28692\sin(2\pi \times 849.4t)] \times 10^{-3} \end{pmatrix} \tag{62}
$$

Combining Equation (61) and Equation (62) yields an approximate analytical expression for the displacement response of the system as:

$$
\mathbf{x}(t) = \dot{u} \begin{pmatrix} [0.6336\sin(2\pi \times 94.1t) + 0.00044\sin(2\pi \times 441.3t) + 1.7983\sin(2\pi \times 849.4t)] \times 10^{-4} \\ [|1.32\sin(2\pi \times 94.1t)| - |0.3801\sin(2\pi \times 441.3t)| + |0.0492\sin(2\pi \times 849.4t)|] \times 10^{-5} \\ [-2.6\sin(2\pi \times 94.1t) + 0.0000068\sin(2\pi \times 441.3t) + 0.28692\sin(2\pi \times 849.4t)] \times 10^{-3} \end{pmatrix} \tag{63}
$$

When $\dot{u} = 0.5$ m/s, the displacement response of the system obtained by the half-value analytical calculation method and the approximate analytical expression is shown in Figure 14; "-ana" represents the result of the approximate analytical expression, and the lateral displacement and rotation angle of both are basically the same. The two axial displacements are basically the same in magnitude, but the specific values are slightly different, which is due to the fact that when the system enters region 4 from region 2 during the approximate analytical calculation, the system state is assumed to be $\mathbf{x} = (0,0,0), \dot{\mathbf{x}} = (-\dot{u},0,0)$, which is different from the actual situation.

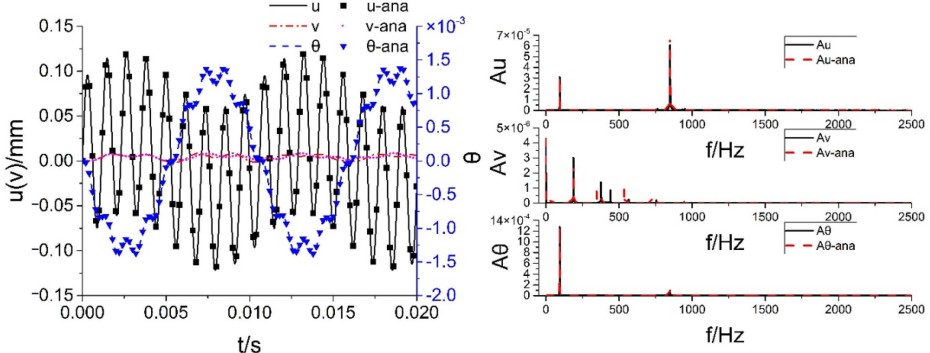

**Figure 14.** Displacement response (**left**) and amplitude frequency (**right**) comparison chart when $\dot{u} = 0.5$ m/s.

The amplitude frequency curves obtained from the displacement responses of the two methods are shown in Figure 14. It can be seen that the displacement response frequencies of the lateral and bending directions of the system obtained by the two methods are basically the same, both being 94.1 Hz and 849.4 Hz, and the magnitude of the displacement response of 441.3 Hz in Equation (63) is very small and can be ignored. The first-order response frequencies of axial displacements are consistent, both being 188 Hz, but the response frequencies of higher orders are significantly different.

In summary, when the lateral impact load is small, the lateral and bending direction displacement response of the system can be solved by Equation (63), and its axial displacement response is small in magnitude and can be neglected.

(2) System response under large lateral impact loads

When the lateral impact load is large (as the resulting lateral initial velocity $\dot{u} \geq 0.8$ m/s), as can be seen from Table 7, the motion time of the system in region 2 and region 4 is significantly reduced. When the lateral impact is large enough, its motion trajectory is spread over all nine regions, and the response is more complex and difficult to analytically calculate, which can only be solved by the semi-numerical analytical calculation method.

The amplitude frequencies of the system are shown in Figure 15, taking $\dot{u} = 1$ m/s and $\dot{u} = 2$ m/s. Compared with the small impact load condition, the amplitude spectrum under a large impact load has more local peaks, and the amplitude spectrum will change with the impact load. When $\dot{u} = 1$ m/s, the first-order response frequency of the system in both lateral and bending directions is 106 Hz, the higher-order response frequency amplitude is smaller, and the axial first-order response frequency is 212 Hz, which is twice that of the lateral and bending directions. When $\dot{u} = 2$ m/s, the first-order response frequency of the system in both lateral and bending directions is 133 Hz, the higher-order response frequency amplitude is significantly higher, and the axial first-order response frequency is about 627 Hz, which is no longer two times that of the lateral and bending directions. Therefore, in the design of the connection structure, attention should be paid to the change in response frequency due to structural nonlinearity to avoid frequency peaks.

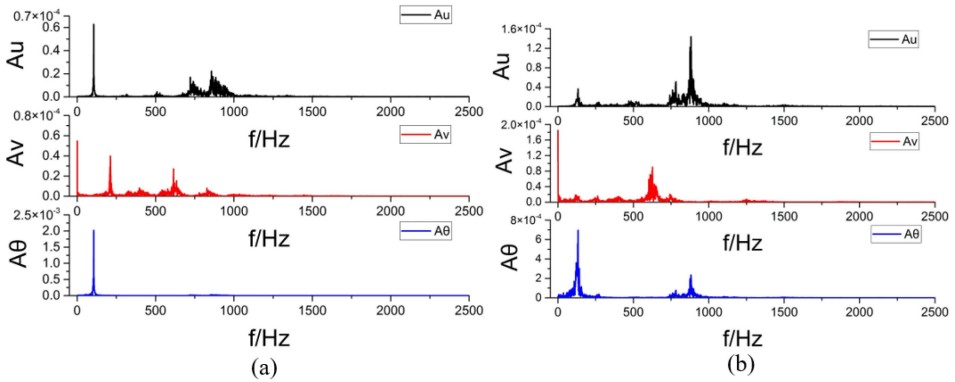

**Figure 15.** Amplitude frequency graph when $\dot{u} = 1$ m/s (**a**) and $\dot{u} = 2$ m/s (**b**).

In fact, in the axial response plot in Figure 15b, there is also a local peak near 212 Hz. However, due to its small amplitude, the response characteristics are not obvious, defining the more pronounced peak at 627 Hz as the axial first-order frequency. This is also the reason for the jump in the axial first-order frequency in Figure 13.

### 3.3.3. System Response under Bending Moment Impact Loading

For a 3-dof trilinear system, the system generates angular velocity under the bending moment impact load, neglecting the time effect of the impact load, and directly assigning the system initial conditions $\mathbf{x}(0) = (0,0,0), \dot{\mathbf{x}}(0) = (0,0,\dot{\theta})$, where $\dot{\theta} \neq 0$; the system response is calculated by the semi-numerical value resolution method.

The displacement response of the system was intercepted within 1 s, and the motion duration of the system in each division was counted, as shown in Table A5. The system response was very sensitive to the magnitude of the impact load, and the first-order response frequencies of each degree of freedom of the system under different lateral impact conditions were obtained by doing the fast Fourier transform on the system response, as shown in Figure 16.

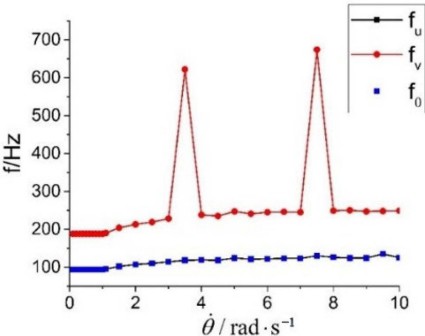

**Figure 16.** First-order response frequency of the system under bending moment impact load.

Since the top side of the cabin is the free end and the bottom side of the cabin is the bound end, the lateral motion of the bottom side always causes the cabin to rotate, and the first-order response frequencies of the system in the lateral and bending directions are exactly the same. Due to the symmetry of the motion, the first-order response frequency in the axial direction is twice as high as that in the lateral direction. However, in specific cases, the amplitude of this multiplier frequency in the axial direction is small, and its first-order frequency migrates to high frequencies. Under small impact loads (i.e., the angular velocity generated is less than or equal to 1 rad/s), the displacement response frequency of the system does not vary with the load, and the first-order displacement response frequencies in the lateral, axial, and bending directions remain at 94 Hz, 188 Hz, and 94 Hz, respectively.

From Table A5, it can be seen that the system mainly moves in regions 2 and 4 under small impact loads, and the motion time in region 5 is short and can be neglected in the approximate analytical calculation. Substituting the vibration type, frequency, and initial conditions of the system into Equation (46), the response of the system in region 4 is:

$$\mathbf{x}_4(t) = -\dot{u}\begin{pmatrix}[-3.948\sin(2\pi\times94.1t) - 0.002\sin(2\pi\times441.3t) + 0.438\sin(2\pi\times849.4t)]\times10^{-5}\\ [8.237\sin(2\pi\times94.1t) - 1.732\sin(2\pi\times441.3t) - 0.012\sin(2\pi\times849.4t)]\times10^{-6}\\ [1.6\sin(2\pi\times94.1t) - 0.000003\sin(2\pi\times441.3t) + 0.007\sin(2\pi\times849.4t)]\times10^{-3}\end{pmatrix} \tag{64}$$

When the system enters region 2 from region 4, the initial condition is approximated by $\mathbf{x}(0) = (0,0,0), \dot{\mathbf{x}}(0) = (0,0,-\dot{\theta})$, then the response in region 2 is:

$$\mathbf{x}_2(t) = \dot{u}\begin{pmatrix}[-3.948\sin(2\pi\times94.1t) - 0.002\sin(2\pi\times441.3t) + 0.438\sin(2\pi\times849.4t)]\times10^{-5}\\ [-8.237\sin(2\pi\times94.1t) + 1.732\sin(2\pi\times441.3t) + 0.012\sin(2\pi\times849.4t)]\times10^{-6}\\ [1.6\sin(2\pi\times94.1t) - 0.000003\sin(2\pi\times441.3t) + 0.007\sin(2\pi\times849.4t)]\times10^{-3}\end{pmatrix} \tag{65}$$

Combining Equations (64) and (65) yields an approximate analytical expression for the displacement response of the system as:

$$\mathbf{x}(t) = \dot{u}\begin{pmatrix}[-3.948\sin(2\pi\times94.1t) - 0.002\sin(2\pi\times441.3t) + 0.438\sin(2\pi\times849.4t)]\times10^{-5}\\ |-8.237\sin(2\pi\times94.1t) + 1.732\sin(2\pi\times441.3t) + 0.012\sin(2\pi\times849.4t)|\times10^{-6}\\ [1.6\sin(2\pi\times94.1t) - 0.000003\sin(2\pi\times441.3t) + 0.007\sin(2\pi\times849.4t)]\times10^{-3}\end{pmatrix} \tag{66}$$

The displacement responses and amplitude frequencies of the half-value semi-analytic calculation method and the approximate analytical method are shown in Figure 17. It can be seen that the displacement responses and amplitude frequencies of the system in the lateral and bending directions obtained by the two methods are basically the same. The axial displacement response is consistent in the initial stage, but when the system enters region 2 from region 4, the approximate analytical method ignores the small amount of system displacement and velocity, which leads to the difference in the later stage.

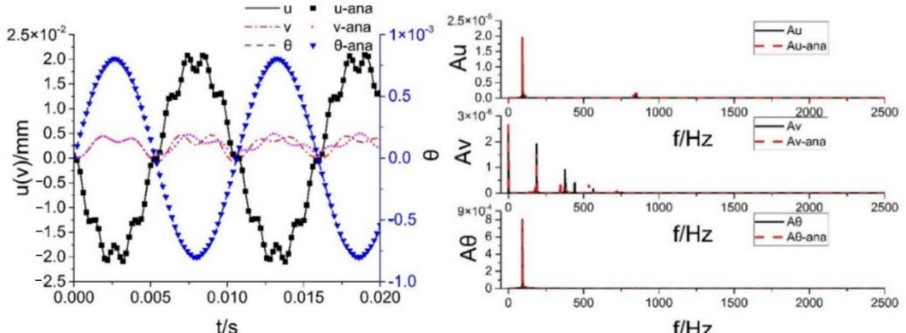

**Figure 17.** Displacement response (left) and amplitude frequency (right) comparison chart when $\dot{\theta} = 0.5 \, \text{rad/s}$.

When the bending impact load is large (i.e., the resulting angular velocity is greater than or equal to 1.1 rad/s), the system motion is no longer confined to regions 2 and 4, and it is difficult to analytically calculate the system response because the motion state of the system cannot be judged when the motion region changes. The displacement response and the amplitude frequency of the system at the initial angular velocity are shown in Figure 18. It can be seen that the response frequency of the system changes slightly under different impact loads, especially the axial displacement response, whose first-order response frequency corresponds to a significant decrease in amplitude under a specific impact load. Due to the nonlinearity of the system, the influence of load magnitude should be considered when analyzing its dynamic characteristics.

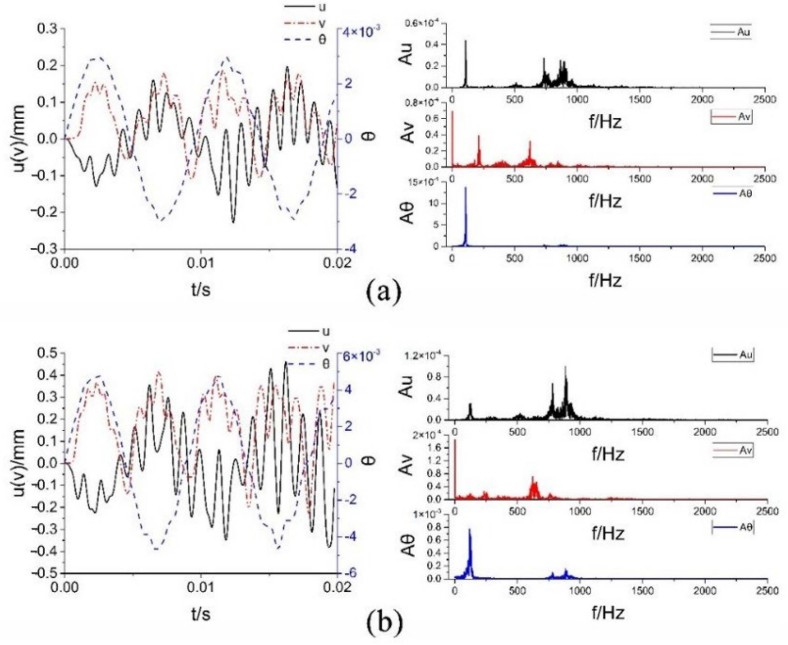

**Figure 18.** Displacement response and amplitude frequency diagram of the system. (**a**) $\dot{\theta} = 2 \, \text{rad/s}$, (**b**) $\dot{\theta} = 3.5 \, \text{rad/s}$.

*3.4. Response under Simple Harmonic Excitation of Sunk Screw Connection Structure*

The parameters of the simplified 3-dof trilinear dynamical system with sunk screw connection structure are shown in Table A3. The response of the system under simple harmonic excitation was analyzed. A simple harmonic excitation in the bending direction can be described as:

$$M = M_0 \sin(2\pi f t) \tag{67}$$

where $M_0$ is the simple harmonic excitation amplitude, $f$ is the simple harmonic excitation frequency, and the unit of $M$ is N·m.

Taking $M_0$ as equal to 1000 N·m and 5000 N·m, respectively, the maximum value of mechanical energy of the system under different bending moment excitation frequencies was obtained by using the half-value analytical calculation method, as shown in Figure 19.

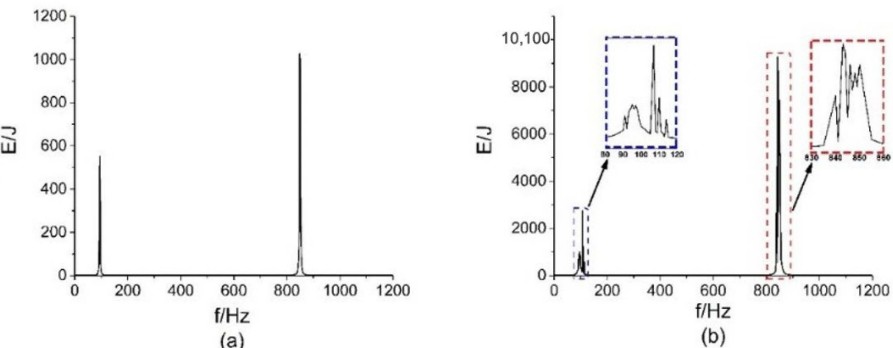

**Figure 19.** Maximum mechanical energy of the system at different excitation frequencies. (**a**) $M_0 = 1000 \, \text{N} \cdot \text{m}$, (**b**) $M_0 = 5000 \, \text{N} \cdot \text{m}$.

It can be found that when the excitation load amplitude is small, the system energy peaks at the excitation frequency of 95 Hz and 848 Hz; when the excitation load amplitude is large, the peak frequency of the system mechanical energy is shifted, the peak range is expanded, and the phenomenon of multi-peak resonance is presented.

## 4. Conclusions

In this study, the stiffness of the sunk screw connection structure was studied by finite element analysis, and it was found that the axial stiffness of the structure was in trilinear form, and the lateral stiffness was in linear form when the assembly gap was considered. The three linear segments of the axial stiffness represented the tensile stiffness, gap stiffness, and compression stiffness, and an empirical formula for stiffness calculation with a certain range of applicability was proposed. It can be used to quickly calculate the connection surface stiffness at the early stages of program design. On this basis, a 3-dof trilinear dynamic model was proposed to study the dynamics of the sunk screw connection structure for the stiffness characteristics of the sunk screw connection structure. The connection surface was simplified into two axial trilinear springs and one lateral linear spring. In order to facilitate the analytical calculation, the system motion was divided into nine motion regions with the linear turning point as the dividing point. The equations of motion in each motion region were derived separately, so the motion of the system in each region could be fully resolved.

The dynamics of a 3-dof trilinear system under impact loading, as well as simple harmonic loading, was investigated by means of a half-numerical analysis method, which has been determined as suitable for most cases where the section size of the cabin structure is much larger than the size of the sunk screw. The motion in this region was analytically calculated by determining the initial motion state of the system and the time in this region.

Under the impact load, the time effect of the impact load was neglected and the initial axial, lateral, or bending direction initial velocity of the system was given. It was found that the response frequency of the system remained constant under smaller impact loads, and the response frequency, as well as the motion response of the system, could be solved by an approximate analytical expression. This was because the main motion region of the system was predictable, and the motion time in other regions was very short and did not significantly affect the overall motion of the system.

When the impact load was large, the full process motion response of the system could not be approximately resolved, and the response frequency fluctuated due to the difficulty in predicting the change of the system motion region and the large influence of the motion

within each region on the overall motion of the system, which reflected the sensitivity of the nonlinear system to the magnitude of the impact load. On the other hand, under the axial impact load, the system produced only axial motion, but applying the impact load in the lateral direction, as well as in the bending moment direction, led to a coupling of the motion of the three degrees of freedom.

Under the bending moment simple harmonic excitation, when the excitation load amplitude was small, the system energy frequency curve showed a good single-peak characteristic; however, when the excitation load amplitude was large, the system energy peak frequency was shifted and showed the phenomenon of multi-peak resonance in a certain range.

This study presented a simplified model of the 3-dof trilinear dynamic model of the sunk screw connection structure, and investigated the dynamics of the system under impact loading and simple harmonic excitation by dividing the motion region and using the half-value calculation method. It revealed the phenomenon that the system motion degrees of freedom were coupled with each other, and the dynamics were sensitive to the load magnitude. This trilinear dynamic model can be further applied to simulate the dynamics of the rocket body to obtain accurate dynamics, which could provide a basis for mastering the key dynamics modeling parameters and designing the dynamic characteristics at the early stage of rocket body design.

**Author Contributions:** Conceptualization, X.Z. and X.L.; methodology, X.L. and C.L.; software, X.L. and C.L.; validation, Z.W., R.W. and X.L.; formal analysis, R.W.; investigation, C.L.; resources, W.W.; data curation, R.W. and W.W.; writing—original draft preparation, R.W.; writing—review and editing, X.Z. and X.L.; visualization, X.Z.; supervision, X.L.; project administration, X.Z.; funding acquisition, X.Z. All authors have read and agreed to the published version of the manuscript.

**Funding:** This research was supported by the National Natural Science Foundation of China [grant number 11972069] and Aeronautical Science Foundation of China [grant number 2017ZA51009].

**Institutional Review Board Statement:** Not applicable.

**Data Availability Statement:** Some or all data, models, or code that support the findings of this study are available from the corresponding author upon reasonable request.

**Conflicts of Interest:** The authors declare that they have no conflict of interest.

**Abbreviations**

$k_{s+}$ = Tensile stiffness (N/m)
$k_{s0}$ = Compression stiffness with clearance (N/m)
$k_{s-}$ = Compression stiffness without clearance (N/m)
$\delta$ = Deformation of sunk screw (mm)
$F$ = Axial load (kN)
$E$ = Elastic modulus of screw material (GPa)
$E_f$ = Elastic modulus of flange material (GPa)
$a_{sX}$ = Correction factor in X direction
$a_{fw}$ = Deformation correction coffcients of outer flage
$a_{fn}$ = Deformation correction coffcients of inner flage
$G_f$ = Shear modulus of flange material (GPa)
$f$ = Response frequency (Hz)

## Appendix A

**Table A1.** Deformation and correction coefficient of compression section with gap.

| Parameter | $\delta_{sX}$/mm | $\delta_{sY}$/mm | $\delta_{fw}$/mm | $\delta_{fn}$/mm | $\delta$/mm | $k_{s0}$/N·m$^{-1}$ |
|---|---|---|---|---|---|---|
| FEM | 0.02374 | 0.02794 | 0.00361 | 0.00152 | 0.04227 | $2.839 \times 10^9$ |
| Results | 0.02372 | 0.02799 | 0.00362 | 0.00153 | 0.04228 | $2.838 \times 10^9$ |
| Correction factor | $a_{sX0}$ | $a_{sY0}$ | $a_{fw0}$ | $a_{fn0}$ | $a_{s0}$ | |
| Value | 2.65 | 1.18 | 1.25 | 0.92 | 0.718 | |

**Table A2.** Spring state and stiffness matrix for each region.

| Region | $[\delta_1,\delta_2]$ | $k_{s1}^*, k_{s2}^*$ | Stiffness Matrix $K_{23}$ | $Q^*$ |
|---|---|---|---|---|
| 1 | $\delta_1 \geq 0$<br>$\delta_2 \geq 0$ | $k_{s1}^* = k_{s+}$<br>$k_{s2}^* = k_{s+}$ | $\begin{bmatrix} 2k_{s+} & 0 \\ 0 & \frac{b^2}{2}k_{s+} \end{bmatrix}$ | $\begin{bmatrix} 0 \\ 0 \\ 0 \end{bmatrix}$ |
| 2 | $\delta_1 \geq 0$<br>$\Delta < \delta_2 < 0$ | $k_{s1}^* = k_{s+}$<br>$k_{s2}^* = k_{s0}$ | $\begin{bmatrix} k_{s+} + k_{s0} & \frac{b}{2}(k_{s0} - k_{s+}) \\ \frac{b}{2}(k_{s0} - k_{s+}) & \frac{b^2}{4}(k_{s+} + k_{s0}) \end{bmatrix}$ | $\begin{bmatrix} 0 \\ 0 \\ 0 \end{bmatrix}$ |
| 3 | $\delta_1 \geq 0$<br>$\delta_2 \leq \Delta$ | $k_{s1}^* = k_{s+}$<br>$k_{s2}^* = k_{s-}$ | $\begin{bmatrix} k_{s+} + k_{s-} & \frac{b}{2}(k_{s-} - k_{s+}) \\ \frac{b}{2}(k_{s-} - k_{s+}) & \frac{b^2}{4}(k_{s+} + k_{s-}) \end{bmatrix}$ | $\begin{bmatrix} 0 \\ \Delta(k_{s-} - k_{s0}) \\ \frac{b}{2}\Delta(k_{s-} - k_{s0}) \end{bmatrix}$ |
| 4 | $\Delta < \delta_1 < 0$<br>$\delta_2 \geq 0$ | $k_{s1}^* = k_{s0}$<br>$k_{s2}^* = k_{s+}$ | $\begin{bmatrix} k_{s+} + k_{s-} & \frac{b}{2}(k_{s-} - k_{s+}) \\ \frac{b}{2}(k_{s-} - k_{s+}) & \frac{b^2}{4}(k_{s+} + k_{s-}) \end{bmatrix}$ | $\begin{bmatrix} 0 \\ 0 \\ 0 \end{bmatrix}$ |
| 5 | $\Delta < \delta_1 < 0$<br>$\Delta < \delta_2 < 0$ | $k_{s1}^* = k_{s0}$<br>$k_{s2}^* = k_{s0}$ | $\begin{bmatrix} 2k_{s0} & 0 \\ 0 & \frac{b^2}{2}k_{s0} \end{bmatrix}$ | $\begin{bmatrix} 0 \\ 0 \\ 0 \end{bmatrix}$ |
| 6 | $\Delta < \delta_1 < 0$<br>$\delta_2 \leq \Delta$ | $k_{s1}^* = k_{s0}$<br>$k_{s2}^* = k_{s-}$ | $\begin{bmatrix} k_{s0} + k_{s-} & \frac{b}{2}(k_{s-} - k_{s0}) \\ \frac{b}{2}(k_{s-} - k_{s0}) & \frac{b^2}{4}(k_{s0} + k_{s-}) \end{bmatrix}$ | $\begin{bmatrix} 0 \\ \Delta(k_{s-} - k_{s0}) \\ \frac{b}{2}\Delta(k_{s-} - k_{s0}) \end{bmatrix}$ |
| 7 | $\delta_1 \leq \Delta$<br>$\delta_2 \geq 0$ | $k_{s1}^* = k_{s-}$<br>$k_{s2}^* = k_{s+}$ | $\begin{bmatrix} k_{s+} + k_{s-} & \frac{b}{2}(k_{s+} - k_{s-}) \\ \frac{b}{2}(k_{s+} - k_{s-}) & \frac{b^2}{4}(k_{s+} + k_{s-}) \end{bmatrix}$ | $\begin{bmatrix} 0 \\ \Delta(k_{s-} - k_{s0}) \\ \frac{b}{2}\Delta(k_{s0} - k_{s-}) \end{bmatrix}$ |
| 8 | $\delta_1 \leq \Delta$<br>$\Delta < \delta_2 < 0$ | $k_{s1}^* = k_{s-}$<br>$k_{s2}^* = k_{s0}$ | $\begin{bmatrix} k_{s0} + k_{s-} & \frac{b}{2}(k_{s0} - k_{s-}) \\ \frac{b}{2}(k_{s0} - k_{s-}) & \frac{b^2}{4}(k_{s0} + k_{s-}) \end{bmatrix}$ | $\begin{bmatrix} 0 \\ \Delta(k_{s-} - k_{s0}) \\ \frac{b}{2}\Delta(k_{s0} - k_{s-}) \end{bmatrix}$ |
| 9 | $\delta_1 \leq \Delta$<br>$\delta_2 \leq \Delta$ | $k_{s1}^* = k_{s-}$<br>$k_{s2}^* = k_{s-}$ | $\begin{bmatrix} 2k_{s-} & 0 \\ 0 & \frac{b^2}{2}k_{s-} \end{bmatrix}$ | $\begin{bmatrix} 0 \\ 2\Delta(k_{s-} - k_{s0}) \\ 0 \end{bmatrix}$ |

**Table A3.** Typical sunk screw connection system parameters.

| Parameters | $M$/kg | $J$/kg·m$^2$ | $b$/m | $r$/m | $\Delta$/m |
|---|---|---|---|---|---|
| Value | 80 | 6.2 | 0.238 | 0.47 | 0.0002 |
| Parameters | $k_{s+}$/N·m$^{-1}$ | $k_{s0}$/N·m$^{-1}$ | $k_{s-}$/N·m$^{-1}$ | $k_r$/N·m$^{-1}$ | |
| Value | $2.95 \times 108$ | $3.2 \times 108$ | $7.25 \times 109$ | $5.7 \times 108$ | |

**Table A4.** Response frequency and vibration pattern of a typical sunk screw connection system.

| Zone | Frequency /Hz | | | Mode of Vibration | | |
|---|---|---|---|---|---|---|
| | $f_1$ | $f_2$ | $f_3$ | $A^{(1)}$ | $A^{(2)}$ | $A^{(3)}$ |
| 1 | 92.3 | 432.2 | 848.8 | $(0.046, 0, -1.974)^{\mathrm{T}}$ | $(0, -1.118, 0)^{\mathrm{T}}$ | $(2.193, 0, 3.498)^{\mathrm{T}}$ |
| 2 | 94.1 | 441.3 | 849.4 | $(0.048, 0.01, -1.971)^{\mathrm{T}}$ | $(0.013, -1.118, 0.002)^{\mathrm{T}}$ | $(2.193, 0.006, 3.499)^{\mathrm{T}}$ |
| 3 | 123.8 | 802.7 | 1718.4 | $(-0.082, -0.208, 1.880)^{\mathrm{T}}$ | $(2.001, -0.456, 3.065)^{\mathrm{T}}$ | $(0.895, 1, 1.788)^{\mathrm{T}}$ |
| 4 | 94.1 | 441.3 | 849.4 | $(-0.048, 0.01, 1.971)^{\mathrm{T}}$ | $(0.013, 1.118, 0.002)^{\mathrm{T}}$ | $(2.193, -0.006, 3.499)^{\mathrm{T}}$ |
| 5 | 96.0 | 450.2 | 850.0 | $(0.050, 0, -1.968)^{\mathrm{T}}$ | $(0, -1.118, 0)^{\mathrm{T}}$ | $(2.193, 0, 3.501)^{\mathrm{T}}$ |
| 6 | 128.4 | 805.4 | 1719.5 | $(-0.088, -0.205, 1.871)^{\mathrm{T}}$ | $(2.003, -0.454, 3.075)^{\mathrm{T}}$ | $(0.891, 1.001, 1.781)^{\mathrm{T}}$ |
| 7 | 123.8 | 802.7 | 1718.4 | $(0.082, -0.208, -1.880)^{\mathrm{T}}$ | $(2.001, 0.456, 3.065)^{\mathrm{T}}$ | $(0.895, -1, 1.788)^{\mathrm{T}}$ |
| 8 | 128.4 | 805.4 | 1719.5 | $(0.088, -0.205, -1.871)^{\mathrm{T}}$ | $(2.003, 0.454, 3.075)^{\mathrm{T}}$ | $(0.891, -1.001, 1.781)^{\mathrm{T}}$ |
| 9 | 327.8 | 1205.1 | 2179.3 | $(0.647, 0, -0.936)^{\mathrm{T}}$ | $(2.096, 0, 3.906)^{\mathrm{T}}$ | $(0, -1.118, 0)^{\mathrm{T}}$ |

**Table A5.** Movement time of each partition of the system under bending moment impact.

| $\dot\theta$/rad·s$^{-1}$ | $t_1$/s | $t_2$/s | $t_3$/s | $t_4$/s | $t_5$/s | $t_6$/s | $t_7$/s | $t_8$/s | $t_9$/s |
|---|---|---|---|---|---|---|---|---|---|
| 0.1 | 0 | 0.4979 | 0 | 0.4990 | 0.0031 | 0 | 0 | 0 | 0 |
| 0.2 | 0 | 0.4979 | 0 | 0.4990 | 0.0031 | 0 | 0 | 0 | 0 |
| 0.4 | 0 | 0.4979 | 0 | 0.4990 | 0.0031 | 0 | 0 | 0 | 0 |
| 0.6 | 0 | 0.4979 | 0 | 0.4990 | 0.0031 | 0 | 0 | 0 | 0 |
| 0.8 | 0 | 0.4979 | 0 | 0.4990 | 0.0031 | 0 | 0 | 0 | 0 |
| 1 | 0 | 0.4979 | 0 | 0.4990 | 0.0031 | 0 | 0 | 0 | 0 |
| 1.1 | 0.0297 | 0.4563 | 0.0237 | 0.4591 | 0.0077 | 0 | 0.0235 | 0 | 0 |
| 1.5 | 0.0315 | 0.3874 | 0.0698 | 0.3845 | 0.0586 | 0 | 0.0682 | 0 | 0 |
| 2 | 0.1128 | 0.3097 | 0.0952 | 0.3106 | 0.0638 | 0.0070 | 0.0961 | 0.0046 | 0.0002 |
| 3 | 0.2518 | 0.1962 | 0.1289 | 0.2017 | 0.0537 | 0.0182 | 0.1308 | 0.0150 | 0.0036 |

Here we give a proof of Equation (45).

The solution of the equation of motion within the region can be set as

$$\mathbf{x} = \mathbf{A}\sin(\omega t + \phi) \tag{A1}$$

where A is the displacement amplitude vector, $\omega$ is the vibration frequency, and $\phi$ is the phase angle. Substituting this into Equation (45) yields:

$$\left(\mathbf{K} - \omega^2\mathbf{M}\right)\mathbf{A} = 0 \tag{A2}$$

The characteristic equation of the system is:

$$\left|\mathbf{K} - \omega^2\mathbf{M}\right| = 0 \tag{A3}$$

By solving this equation, we can obtain $n$ positive real roots $\omega_i(i = 1, 2, \cdots, n)$, which are the $n$ intrinsic frequencies of the system. Then Equation (A2) can be written as

$$\mathbf{E}^{(i)}\mathbf{A}^{(i)} = 0 \tag{A4}$$

$$\mathbf{E}^{(i)} = \mathbf{K} - \omega_i^2\mathbf{M} \tag{A5}$$

$\mathbf{A}^{(i)}$ is the eigenvector corresponding to the eigenvalues $\omega_i^2$.

When the characteristic Equation (A3) has no repeated roots, only one of the $n$ equations of Equation (A4) is not independent and the first element of the characteristic vector can be taken as the unit magnitude.

$$\mathbf{A}^{(i)} = [1 \ a_2^{(i)} \ a_3^{(i)} \ \ldots \ a_n^{(i)}]^{\mathrm{T}} \tag{A6}$$

Expanding and blocking Equation (A4), we obtain:

$$\begin{bmatrix} e_{11}^{(i)} & e_{12}^{(i)} & \cdots & e_{1n}^{(i)} \\ e_{21}^{(i)} & e_{22}^{(i)} & \cdots & e_{2n}^{(i)} \\ \vdots & \vdots & \ddots & \vdots \\ e_{n1}^{(i)} & e_{n2}^{(i)} & \cdots & e_{nn}^{(i)} \end{bmatrix} \begin{Bmatrix} 1 \\ a_2^{(i)} \\ \vdots \\ a_n^{(i)} \end{Bmatrix} = \begin{Bmatrix} 0 \\ 0 \\ \vdots \\ 0 \end{Bmatrix} \tag{A7}$$

$$\begin{bmatrix} e_{11}^{(i)} & \mathbf{E}_{12}^{(i)} \\ \mathbf{E}_{21}^{(i)} & \mathbf{E}_{22}^{(i)} \end{bmatrix} \begin{Bmatrix} 1 \\ \mathbf{A}_2^{(i)} \end{Bmatrix} = \begin{Bmatrix} 0 \\ \mathbf{0} \end{Bmatrix} \tag{A8}$$

$$\mathbf{A}_2^{(i)} = -\left[\mathbf{E}_{22}^{(i)}\right]^{-1}\mathbf{E}_{21}^{(i)} \tag{A9}$$

At this point, the i-th order feature vector is obtained.

Equation (A2) can be extended to the form of the vibration and spectral matrices:

$$\mathbf{K\Phi=M\Phi\Lambda} \tag{A10}$$

$$\boldsymbol{\Phi} = \begin{bmatrix} \mathbf{A}^{(1)} & \mathbf{A}^{(2)} & \cdots & \mathbf{A}^{(n)} \end{bmatrix} \tag{A11}$$

$$\boldsymbol{\Lambda} = \begin{bmatrix} \omega_1^2 & & & \\ & \omega_2^2 & & \\ & & \ddots & \\ & & & \omega_n^2 \end{bmatrix} \tag{A12}$$

Substituting the ith-order intrinsic frequency $\omega_i$ and the displacement amplitude vector $\mathbf{A} = a_i \mathbf{A}^{(i)}$ into Equation (A1), we obtain

$$\mathbf{x}_i = a_i \mathbf{A}^{(i)} \sin(\omega_i t + \phi_i) \tag{A13}$$

This is the i-th order vibration pattern.

Taking the labels of Equation (A13) from 1 to $n$ and superimposing them, the undamped free vibration form of the system is obtained as

$$\mathbf{x}(t) = \sum_{i=1}^{n} a_i \mathbf{A}^{(i)} \sin(\omega_i t + \phi_i) = \sum_{i=1}^{n} \mathbf{A}^{(i)} (B_i \cos \omega_i t + C_i \sin \omega_i t) \tag{A14}$$

The solution of the free vibration is obtained by giving 2 $n$ initial conditions of the system. Suppose the initial displacement and initial velocity of the system are

$$\mathbf{x}(0) = \mathbf{x}_0, \dot{\mathbf{x}}(0) = \dot{\mathbf{x}}_0 \tag{A15}$$

Substituting Equation (A14) into Equation (A15) yields

$$\mathbf{x}(0) = \sum_{i=1}^{n} \mathbf{A}^{(i)} B_i, \dot{\mathbf{x}}(0) = \sum_{i=1}^{n} \mathbf{A}^{(i)} \omega_i C_i \tag{A16}$$

According to the orthogonality of the vibration pattern there are

$$\begin{cases} \mathbf{A}^{(i)T} \mathbf{M} \mathbf{A}^{(j)} = 0, (i \neq j) \\ \mathbf{A}^{(i)T} \mathbf{K} \mathbf{A}^{(j)} = 0, (i \neq j) \\ \mathbf{A}^{(i)T} \mathbf{M} \mathbf{A}^{(i)} = M_i \\ \mathbf{A}^{(i)T} \mathbf{K} \mathbf{A}^{(i)} = K_i \end{cases} \tag{A17}$$

Multiplying Equation (A16) left by $\mathbf{A}^{(i)T}\mathbf{M}$ and $\mathbf{A}^{(i)T}\mathbf{K}$ respectively, we get

$$B_i = \frac{\mathbf{A}^{(i)T} \mathbf{M} \mathbf{x}(0)}{M_i}, C_i = \frac{\mathbf{A}^{(i)T} \mathbf{M} \dot{\mathbf{x}}(0)}{\omega_i M_i} \tag{A18}$$

Thus, the undamped free vibration form of the system can be obtained as

$$\mathbf{x}(t) = \sum_{i=1}^{n} \mathbf{A}^{(i)} \left( \frac{\mathbf{A}^{(i)T} \mathbf{M} \mathbf{x}(0)}{M_i} \cos \omega_i t + \frac{\mathbf{A}^{(i)T} \mathbf{M} \dot{\mathbf{x}}(0)}{\omega_i M_i} \sin \omega_i t \right) \tag{A19}$$

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
