# Peer review of "Dynamics Modeling and Characterization of Sunk Screw Connection Structure in Small Rockets"

_aerospace, doi:10.3390/aerospace9110648_

Round 1
Reviewer 1 Report
Thanks for submitting the paper.
The English language and style are poor, spell check required at certain places.
More content can be added to explain the significance of the work.
More details on FEA simulation can be added, including the stress results.
Section-2, Please explain the simplified 2D FE model results as compared to fully 3D model analysis.
Provides more details about the FE analysis, for example, what commercial FEA tool used, the element type, size of the model.
Have you considered the nonlinearity in your FE analysis?
Looks like, it is a linear elastic static analysis to calculate the stiffness in X- and Y-directions. Have you considered a combination of multiple load cases to capture the design envelope for multi-axial loading scenarios?
Section-2.2, could you emphasize and provide more explanation to justify “axial stiffness can be equated to a trilinear spring and its lateral stiffness to a linear spring”?
Reviewer 2 Report
The paper address issue of modelling of the sunk screw connection. The authors provided brief state of the art and justify well why their paper is needed and why it is novel. The authors starts from finite element model and on the basis of its response they propose a simplified dynamics model that can be useful in practice. Analysis of results is comprehensive. However, in my
opinion authors should write more about their basic FE model since its results is it starting point. What is more, in my opinion some more comments on verification/ validation of their solutions should be made.
1) In my opinion some details of FEA model should be revealed.
a) Type of analysis: statics, dynamics, if dynamics what algorithm was used to solve it.
b) Type of finite element
c) Contact model in each model case and friction model.
d) Boundary conditions in the model (I suggest to draw them with loading in each model case on the additional Figure)
e) Material model
f) Have you performed convergence analysis?
g) Do you use your own FE code or commercial software? Please write where did you performed computations.
2) The authors published earlier very detailed model. Li, Xiaogang, et al. "Study on the behavior of sunk screw connection between cabins in the rocket." Acta Astronautica 166 (2020): 199-208.
Could you refer to this model? Do you use in this paper FEA model that was validated with experiment? If yes, it would be beneficial to mentioned. If not, do you plan to validate your
approach by comparison with experiment?
3) You start from FEA model to derive your analytical approach. Then you use analytical and half-value analytical approach. Could you comment on verification and validation of your analytical
approach? Would be possible to compare this results with experiment or results obtained by validated model?
4) Minor remarks
a) In some places (i guess places of citation) some Chinese (?) marks appear instead of reference to Tables? I suppose that this a mistake.
b) Section 2.1.1, line 156: “ 60 kN, 120kN and 180 kN, respectively”: it is not clear respectively to what this values of load are given, to what parts of structures?
c) Fig 2. I suggest to add in the caption unit of the displacement
d) Line 239 “Table 120. kN and the results...” please revise this sentence, I suppose it is not correct ( “Table 120. kN”)
e) Please revise symbols of mathematical variables in the entire manuscript . E.g Sometimes you use k0 without subscript in the text (e.g line 296) , and sometimes with subscript in the questions. Sometimes symbols in the text are not italic but in the equations they are.

Round 2
Reviewer 1 Report
Thanks for including additional details, as suggested in the previous review.
Reviewer 2 Report
Thank you making revision. New version of the paper is in my opinion better. I have only few comments:
1) Quality of new Fig 9 is low. I cannot see what is on each axis of the plots. I recommend to make the graph lager
2) You imposed higher elastic modulus of the cabin material in 3.2. I understand the need to be closer to the other model. However, could you comment in the text how assuming realistic value would affect the behaviour? Is your simplified model close enough to the reality? I suggest to summarise the limitations of your new model.
3) Last paragraph of instruction, first sentence “This paper proposes an empirical formula for stiffness calculation with a certain range of applicability, which can be used to quickly calculate the joint surface stiffness at the early stage of rocket scheme design, and to obtain the joint surface stiffness by accurate finite element model or static test at the detailed design stage. “ Is the part of obtaining joint stiffness by accurate finite model or static test should be here? Do you use your empirical formula in detailed FE model and static test? I suggest to clarify this paragraph.
